# Strain-Specific Features of Primary Metabolome Characteristic for Extremotolerant/Extremophilic Cyanobacteria Under Long-Term Storage

**DOI:** 10.3390/ijms26052201

**Published:** 2025-02-28

**Authors:** Tatiana Bilova, Nikita Golushko, Nadezhda Frolova, Alena Soboleva, Svetlana Silinskaia, Anna Khakulova, Anastasia Orlova, Maria Sinetova, Dmitry Los, Andrej Frolov

**Affiliations:** 1Laboratory of Analytical Biochemistry and Biotechnology, K.A. Timiryazev Institute of Plant Physiology Russian Academy of Sciences, 127276 Moscow, Russia; st078415@student.spbu.ru (N.G.); frolovanadja@yandex.ru (N.F.); soboleva@ifr.moscow (A.S.); svetlanasilsv@mail.ru (S.S.); orlova@ifr.moscow (A.O.); 2Department of Plant Physiology and Biochemistry, St. Petersburg State University, 199034 St. Petersburg, Russia; 3Chemical Analysis and Materials Research Core Facility Center, Research Park, St. Petersburg State University, 199034 St. Petersburg, Russia; a.khakulova@spbu.ru; 4Laboratory of Intracellular Regulation, K.A. Timiryazev Institute of Plant Physiology Russian Academy of Sciences, 127276 Moscow, Russia; maria.sinetova@mail.ru (M.S.); losda@ippras.ru (D.L.)

**Keywords:** cyanobacterial strains, extremophiles, haloalkaliphiles and natronophiles, adaptation to extreme conditions, metabolomics, GC-MS, LC-MS, primary metabolites, metabolic pathways

## Abstract

Cyanobacteria isolated from extreme habitats are promising in biotechnology due to their high adaptability to unfavorable environments and their specific natural products. Therefore, these organisms are stored under a reduced light supply in multiple collections worldwide. However, it remains unclear whether these strains maintain constitutively expressed primary metabolome features associated with their unique adaptations. To address this question, a comparative analysis of primary metabolomes of twelve cyanobacterial strains from diverse extreme habitats was performed by a combined GC-MS/LC-MS approach. The results revealed that all these cyanobacterial strains exhibited clear differences in their patterns of primary metabolites. These metabolic differences were more pronounced for the strains originating from ecologically different extreme environments. Extremotolerant terrestrial and freshwater strains contained lower strain-specifically accumulated primary metabolites than extremophilic species from habitats with high salinity and alkalinity. The latter group of strains was highly diverse in amounts of specific primary metabolites. This might indicate essentially different molecular mechanisms and metabolic pathways behind the survival of the microorganisms in saline and alkaline environments. The identified strain-specific metabolites are discussed with respect to the metabolic processes that might impact maintaining the viability of cyanobacteria during their storage and indicate unique adaptations formed in their original extreme habitats.

## 1. Introduction

Cyanobacteria are an ancient group of photosynthetic microorganisms found in diverse terrestrial and water environments, significantly impacting water quality and the functioning of aquatic ecosystems [1]. In addition to their ecological roles, they are essential components of food webs, serving as the only prokaryotic organisms, primary producers performing oxygenic photosynthesis [2]. Another noteworthy attribute of cyanobacteria is their remarkable adaptability to a diverse range of extreme environments such as drought, high or low temperatures, ultraviolet radiation, salinity and alkalinity [3]. Some cyanobacteria, considered extremophilic, have developed unique adaptation strategies during their extensive evolutionary history that allow them to fine-tune their metabolism to ‘extreme’ conditions. It is important to note that ‘extreme’ conditions represent the norm at which adapted organisms are able to function metabolically and biochemically without experiencing a state of stress. Other cyanobacteria considered as extremotolerant have adapted to tolerate a wide range of variation in one or more life-limiting parameters (e.g., temperature variation, water availability) [4].

Different molecular mechanisms mediating cyanobacteria adaptations are known. As they are exposed to UV radiation in their natural habitats, the majority of cyanobacteria developed a few UV-stress mitigation mechanisms: enhanced biosynthesis of carotenoids [5], mycosporine-like amino acids [6], polyphenols [7] and polyamines to protect the photosynthetic apparatus; up-regulation of scavengers for reactive oxygen species (ROS) and antioxidant machinery [8,9]; induction of repairing systems for UV-damaged biopolymers [10]. Additional strategies used by desiccation-tolerant cyanobacteria include the production of extracellular polymeric substances, recruitment of chaperones to maintain protein integrity, activation of ion channels, synthesis of compatible solutes to overcome low water availability [11] and synthesis of the non-reducing sugar trehalose to protect membranes during desiccation [12]. To withstand low temperatures, cyanobacteria are known to produce cold shock proteins, antifreeze proteins and cryoprotectants (e.g., dimethylsulfoniopropionate [13,14]), protecting against freezing damage [15]. The adaptations that cyanobacteria have evolved to survive in saline environments (seawater, hypersaline lakes) include a profound reorganization of regulatory systems. These ensure the functioning of fundamental processes such as the maintenance of osmotic pressure [16,17,18], cellular homeostasis, energy production and the excretion of sodium ions [19]. Indeed, the genomes of the halotolerant cyanobacteria carry clusters of genes encoding proteins involved in the adaptation to high salinity. Interestingly, in some cases, the metabolic adjustment to extreme conditions enables cyanobacteria to produce highly valuable substances with functional importance and potential human health benefits [18]. This makes cyanobacteria a source of natural compounds with characteristic biological properties and a potential candidate for pharmaceuticals.

The current work was focused on comparative profiling primary metabolites of cultivated cyanobacterial strains previously extracted from various extreme environments (high or low temperatures, drought, increased salinity and alkalinity). Primary metabolites, the small molecules involved in major metabolic pathways crucial for energy production, carbon fixation, the synthesis of essential biopolymers and secondary compounds [20], are the major players mediating the diversity of adaptive metabolic responses in cyanobacteria. They appeal to scientific interest for at least two reasons. First, these adaptations hold potential for biotechnological applications due to the production of unique biologically active compounds [21]. Second, the cyanobacterial strains can be used as models to address the molecular basis of their exceptional adaptivity and transfer the knowledge gained for improving the tolerance of crops to adverse conditions [19]. However, despite progress in understanding cyanobacterial adaptations to extreme environments, many aspects remain uncertain. Specifically, the intricate details of the molecular mechanisms and regulatory networks involved in these adaptive processes in changeable environments are not fully elucidated. Additionally, the comprehensive profiles of primary metabolites and their specific roles in enabling cyanobacterial survival under such conditions are still largely unknown [22,23].

To effectively study the primary metabolism of these cyanobacteria, their primary metabolomes need to be addressed. This can be done using modern high-throughput analytical techniques such as gas chromatography–mass spectrometry and liquid chromatography–mass spectrometry (GC-MS and LC-MS). GC-MS analyzes small molecules (e.g., amino acids, sugars) by converting them into volatile thermally stable derivatives through a derivatization process [24]. LC-MS, on the other hand, analyzes small molecules not suitable for GC-MS due to their thermal instability [25]. This combination of methods acquires a comprehensive metabolite profile, addressing the limitations of each technique [26,27,28]. Here, we employed the advanced GC-MS and LC-MS techniques to perform a comparative analysis of the primary metabolomes of twelve cyanobacterial strains. The strains, isolated from different extreme habitats, were cultivated in strain-specific media under conditions suboptimal for growth, which were provided by a reduction in light and the absence of an additional CO_2_ supply. These cultivation conditions guaranteed the long-term maintenance (storage) of the strains in a metabolically active state [29]. The long-term storage of metabolically active cyanobacterial cultures is of great importance in order to guarantee the sustainable availability of their diversity for fundamental and applied research purposes, as well as for the biotechnological production of unique cyanobacterial bioactive compounds [30]. This exploratory study is an attempt to investigate whether strains cultivated under long-term storage conditions retain a set of primary metabolites that may be related to the unique adaptations formed in the specific extreme conditions in which the cyanobacteria originally evolved.

## 2. Results

### 2.1. Characterization of Studied Extremophilic and Extremotolerant Cyanobacteria Strains

The studied strains were isolated from different extreme habitats (Table 1) and were maintained in the IPPAS collection by periodic transfer as unialgal non-axenic cultures for 4–50 years before the present study. The strains of genus *Sodalinema*, IPPAS B-2050, IPPAS B-2037 and IPPAS B-353, are the reference strains of *S. stalii*, *S. orleanskyi* and *S. gerasimenkoae*, respectively, which were described based on a polyphasic approach by Samylina et al. [31]. Other strains were tentatively identified based on their partial 16S rRNA sequences and morphology (Appendix A).

Based on their habitat, the studied strains can be divided into desiccation-tolerant, high- and low-temperature-tolerant, and halophilic, haloalkaliphilic and natronophilic groups. Desiccation-tolerant strains of *Nostoc commune* B-1519 and B-1520 were isolated from the same macrocolony collected from the soil surface. Such macrocolonies are subject to periodic desiccation during the summer period. The thermotolerant freshwater strain *Dolichospermum* sp. IPPAS B-1213 was isolated from a hot spring and has an optimal growth temperature of 38 °C [32]. The cold-tolerant strains *Anabaena* sp. IPPAS B-1535 and *A.* cf. pirinica IPPAS B-1533 were isolated from the Siberian river Yenesei, where the water temperature ranges from 5 to 14 °C in spring–summer period and the flow velocity is up to 2 m s^−1^ [33]. Haloalkaliphilic and natronophilic strains of *Sodalinema orleanskyi* B-2037, *S. gerasimenkoae* B-353, *Limnospira* sp. B-256, B-287, and B-1526, and *Nodularia* sp. B-1529 were isolated from saline–alkaline and soda lakes. Such mineral lakes are characterized by alkaline pH (9–11.5) and high concentrations of Na^+^ (K^+^) and Cl^−^; soda lakes additionally contain bicarbonates and carbonates [34,35]. It is important to note that salinity in some mineral lakes (e.g., soda lakes in the Altai Region, Russia) can exhibit dramatic fluctuations depending on the precipitation–evaporation balance [36]. Halophilic strain *S. stalii* B-2050 was isolated from the North Sea.

**Table 1 ijms-26-02201-t001:** List of investigated extremotolerant and extremophilic cyanobacteria.

Tolerance Group	IPPAS ID	Species Name	Extreme Environment	Storage Conditions for Cyanobacteria Cultivation
Desiccation-tolerant	B-1520	*Nostoc commune*	Macrocolony collected from soil surface, Gorodets village, Kaluga obl., Russia. Heterocystous diazotroph. Resistant to desiccation.	BG-11 medium without nitrogen (pH 7.5) [37], light intensity 50 µmol photons m^−2^ s^−1^, in an orbital shaker at 22 °C for 3 months.
	B-1519	*Nostoc commune*	Macrocolony collected from soil surface, Gorodets village, Kaluga obl., Russia. Heterocystous diazotroph. Resistant to desiccation.	BG-11 medium without nitrogen (pH 7.5) [37], light intensity 50 µmol photons m^−2^ s^−1^, in an orbital shaker at 22 °C for 3 months.
High- and low-temperature-tolerant	B-1213	*Dolichospermum* sp.	Hot springs, Karlovy Vary, Czech Republic. Heterocystous diazotroph. Thermophile.	BG-11 medium without nitrogen (pH 7.5) [37], light intensity 50 µmol photons m^−2^ s^−1^, in an orbital shaker at 22 °C for 8 months.
	B-1533	*Anabaena* cf. pirinica	Yenisei river, Krasnoyarsk, Russia. Cold-tolerant: able to survive at low temperatures (up to 10–11 °C) and high water flow rate. Heterocystous diazotroph.	No. 6 medium without nitrogen (pH 7.2) [38], light intensity 50 µmol photons m^−2^ s^−1^, in an orbital shaker at 22 °C for 3 months.
	B-1535	*Anabaena* sp.	Yenisei river, Krasnoyarsk, Russia. Cold-tolerant: able to survive at low temperatures (up to 10–11 °C) and high water flow rate. Heterocystous diazotroph.	No. 6 medium without nitrogen (pH 7.2) [38], light intensity 50 µmol photons m^−2^ s^−1^, in an orbital shaker at 22 °C for 3 months.
Halophilic, haloalkaliphilic and natronophilic	B-2050	*Sodalinema stalii*	Coastal shoals, Mellum Island, North Sea, Germany. Salinity about 30 g/L. Halophilic.	ASNIII medium (pH 7.5) [39], light intensity 50 µmol photons m^−2^ s^−1^, 27 °C in a growth chamber MLR-352-PE (Panasonic, Osaka, Japan) for 3 weeks, then 22 °C for 2 months.
	B-2037	*Sodalinema orleanskyi*	Salt alkaline lake Eyasi, Tanzania. Haloalkaliphile, natronophile (pH_opt_ 9–10, growth requires 0.2 M NaHCO_3_ in the medium).	S medium (pH 9.0–9.5) [31], light intensity 50 µmol photons m^−2^ s^−1^, 32 °C in a growth chamber MLR-351 (SANYO, Osaka, Japan) for 3 weeks, then 22 °C for 2 months.
	B-353	*Sodalinema gerasimenkoae*	Salt alkaline lake Khilganta, Transbaikal Territory, Russia. Haloalkaliphile, natronophile (pH_opt_ 9–10, growth requires 0.2 M NaHCO_3_ in the medium).	S medium (pH 9.0–9.5) [31], light intensity 50 µmol photons m^−2^ s^−1^, 27 °C in a growth chamber MLR-352-PE (Panasonic, Osaka, Japan) for 3 weeks, then 22 °C for 2 months.
	B-1526	*Limnospira* sp.	Soda Lake Gorchina I, Altai Region, Russia. Haloalkaliphile, natronophile (pH_opt_ 9–10, growth requires 0.2 M NaHCO_3_ in the medium).	Zarrouk medium (pH 9.5) [40], light intensity 50 µmol photons m^−2^ s^−1^, 32 °C in a growth chamber MLR-351 (SANYO, Osaka, Japan) for 6 weeks.
	B-287	*Limnospira* sp.	The origin of the strain is not precisely known. Haloalkaliphile, natronophile (pH_opt_ 9–10, growth requires 2 M NaHCO_3_ in the medium).	Zarrouk medium (pH 9.5) [40], light intensity 50 µmol photons m^−2^ s^−1^, 32 °C in a growth chamber MLR-351 (SANYO, Osaka, Japan) for 6 weeks.
	B-256	*Limnospira* sp.	Bodou Soda Lake, Chad. Haloalkaliphile, natronophile (optimum pH 9–10, growth requires 0.2 M NaHCO_3_ in the medium).	Zarrouk medium (pH 9.5) [40], light intensity 50 µmol photons m^−2^ s^−1^, in a growth chamber MLR-351 (SANYO, Osaka, Japan) 32 °C for 6 weeks.
	B-1529	*Nodularia* sp.	Soda Lake Gorchina I, Altai Region, Russia. Haloalkaliphile, natronophile (pH_opt_ 9–10, growth requires 0.2 M NaHCO_3_ in the medium). Heterocystous diazotroph.	Zarrouk medium without nitrogen (pH 9.5) [40], light intensity 50 µmol photons m^−2^ s^−1^, 27 °C in a growth chamber MLR-352-PE (Panasonic, Japan) for 3 weeks, then 22 °C for 3 weeks.

### 2.2. Characterization of Extremophilic and Extremotolerant Cyanobacteria Polar Metabolite Patterns

The GC-MS-based untargeted analysis of derivatized aq. methanolic extracts of all extremophilic and extremotolerant cyanobacterial strains investigated here allowed the detection of a total of 358 total ion current (TIC) peaks of thermally stable polar metabolite trimethylsilyl (TMS) and methoxy-(MEOX)-TMS derivatives (features) (Appendix A). Out of this number, 75 features were identified by co-elution with authentic standards from an in-house authentic standard library (ihASL). Structural annotation of the other 74 features relied on close similarity in experimentally obtained retention index (RI) and electron ionization mass spectrum (EI-MS) with those GC-MS data available for standard compounds from known spectral libraries, GMD and NIST. The identified metabolites were represented by following classes: 15 organic acids (lactic, oxalic, glycolic, succinic, glyceric, itaconic, fumaric, salicylic, *D*-erythronic, 2-ketoglutaric, citric, isocitric, propanoic, gluconic, mucic), 15 amino acids (Val, Ala, Gly, Tyr, Leu, Ile, Ser, Asp, Glu, Phe, Lys, Trp, Orn, homoserine, *γ*-aminobutyric acid), 11 sugars and derivatives (ribose, xylose, fructose, mannose, galactose, glucose, *N*-acetyl-glucosamine, sucrose, maltose, isomaltose, α,α-trehalose), 8 polyols (glycerol, erythritol, 1-deoxypentitol, arabitol, ribitol, mannitol, sorbitol, *myo*-inositol), 4 sugar phosphates and other organic phosphates (*myo*-inositol phosphate, fructose 6-phosphate, glucose 6-phosphate, methyl-phosphate), 20 fatty acids and derivatives (nonanoic acid, dodecanoic acid, 1-dodecanol, tetradecanoic acid, *n*-pentadecanoic acid, palmitic acid, methyl palmitate, *trans*-9-hexadecenoic acid, *cis*-9-hexadecenoic acid, heptadecanoic acid, *cis*-10-heptadecenoic acid, stearic acid and its methyl ester, *γ*-linolenic acid and its methyl ester, linoleic acid and its methyl, oleic acid, *trans*-9-octadecenoic acid, nonadecan-1-ol) and 29 representatives of other classes. The third group of 100 features with structures that could not be established by similarity with library data was annotated by a presence in their EI-MS fragment ions that might be considered according to Harvey and Vouros [41] as diagnostic for specific chemical classes as follows: monosaccharides and their derivatives (*m*/*z* 204, 217, 319), di-, tri- or oligosaccharides (*m*/*z* 204, 217, 361), amino acids and their derivatives (*m*/*z* 100, 174), phosphate-containing substances (*m*/*z* 299, 315, 357), fatty acids (*m*/*z* 117, 129). Furthermore, 109 features could not be annotated to any chemical class and were categorized as unknowns. Annotations of the unknowns and features related to a specific chemical class as well were assigned with their unique RI.

LC-MS analysis based on the target MRM mode allowed the detection and identification of a total of 183 thermally labile analytes in polar extracts of all investigated extremophilic and extremotolerant cyanobacterial strains. The identified metabolites were representatives of chemical classes such as amino acids and their derivatives (Arg, Gln, Gly, Met, Ser, Thr, Tyr, Phe, Asp, Glu, Trp, citrulline, S-adenosyl-*L*-homocysteine), sugars and their derivatives (sucrose, glucopyranuronic acid, galactopyranuronic acid, galactonic acid/gluconic acid), sugar phosphates and other organic phosphates (21 identified metabolites), nucleotides (32 identified metabolites), coenzyme A thioesters (6 identified metabolites) and other compounds involved in energy metabolic pathways (Appendix A). Importantly, patterns of metabolites detected by LC-MS and GC-MS were partially overlapped for some amino acids, organic acids, carbohydrates and representatives of other chemical classes. Although all these metabolites have been isolated from cyanobacteria, it cannot be excluded, particularly in the case of *trans*-fatty acids, that they may also originate from bacteria that accompany cyanobacterial cultures.

### 2.3. Identification of Strain-Dependent Variability in the Primary Metabolome

The combination of the GC-MS- and LC-MS-originated datasets resulted in a merged matrix consisting of 530 analytes in total. This matrix was subjected to statistical processing to estimate the magnitude of metabolome variance among the studied extremophilic and extremotolerant cyanobacteria strains. PCA built for the first two PCs, explaining a total of 49.7% of the variance, revealed the following regularities in sample clustering (Figure 1):

(1) Quality control samples (QCs) whose metabolite compositions were analyzed by both GC-MS- and LC-MS-based methods displayed a clear cluster separated from all other studied sample groups. The QC cluster was characterized by low in-group variance that serves as validation of the high quality of the data acquisitions performed without any serious technical problems during GC-MS and LC-MS analyses.

(2) The sample metabolomes formed distinct clusters attributed to the studied cyanobacterial strains. Furthermore, the metabolomes of the cyanobacterial strains exhibited additional clustering according to the strain extrema ecological groups. Indeed, in the PCA model, two distinct clusters were formed by desiccation-tolerant strains (B-1520 and B-1519, Figure 1, cluster 1) and temperature-tolerant strains (low-temperature B-1533, B-1535 and high-temperature B-1213, Figure 1, cluster 2). The clustering of haloalkaliphilic and natronophilic cyanobacteria strains (B-2037, B-1529, B-353, B-1526, B-256, B-287, B-2050, Figure 1, cluster 3) was more complicated and could be described by dividing the strains into two subgroups. The first subgroup denoted strains B-2050, B-353, B-1526 and B-2037 (Figure 1, cluster 3.1) without considerable intergroup metabolome variability. Conversely, strains, B-287, B-256 and B-1529 (Figure 1, cluster 3.2) within the second subgroup were characterized by essential intergroup variability. Additionally, the observed difference in distances between the strain clusters in the first and second subgroups indicates distinctions and similarities, respectively, in metabolite patterns in strains in this haloalkaliphilic and natronophilic ecological group.

(3) PCA also showed that cyanobacterial strains with heterocyst-forming ability (heterocystous diazotrophs B-1520, B-1519, B-1213, B-1533, B-1535, B-1529) formed a cluster separated from the other strains without this ability.

(4) An analysis of the PCA loadings plot revealed metabolites such as succinate, sugar (RI2304), trehalose, malate and glucosylglycerol contributing to the observed clustering of studied strain metabolomes (Appendix A).

In the next step of data analysis, a heatmap was created. This method is very effective in drawing attention to strains characterized by rich patterns of accumulated metabolites and assessing potential relationships between the strains (Figure 2). Indeed, the heatmap has clustered a group of three strains (B-1529, B-256, B-287) out of the majority of other cyanobacteria. The strains were assigned to the haloalkaliphilic and natronophilic ecological group and were highlighted by the highest number of accumulated compounds. It is worth remembering that these strains were also selected by PCA as the most distinguished ones (i.e., characterized with the highest intergroup variability) from other strains. Further clustering of the remaining majority of strains resulted in the isolation of the two cold-tolerant strains (B-1533 and B-1535) from a cluster of the remaining seven strains representing groups resistant to desiccation, high temperature, and haloalkaliphilic and natronophilic.

Overall, the obtained results indicated similarities and distinctions in metabolite profiles among the studied cyanobacterial strains. Despite the lengthy period during which cyanobacterial strains have been cultivated under storage conditions, it was observed that metabolomic similarities were more characteristic of strains originating from the same ecological extreme habitats. In contrast, differences in metabolite profiles were more characteristic of strains from ecologically different extreme habitats. Nevertheless, the exception to this regularity was a few strains of the haloalkaliphilic and natronophilic ecological group which demonstrated distinct clustering between themselves and strains of other extreme habitats. Among them, three strains, B-1529, B-256 and B-287, contained the highest number of accumulated metabolites.

### 2.4. Identifying Patterns of Strain Differences in Primary Metabolome Associated with Cyanobacteria Inhabiting Extreme Environments

To elucidate the key metabolites that could potentially be involved in the adaptations formed by cyanobacteria under extreme conditions, a comparative pairwise analysis of the primary metabolome of each of the strains studied was performed with the metabolomes of all other strains. To facilitate the identification of regularities that may be observed in metabolite patterns and be associated with a strain’s extreme habitat, the comparative analysis results are described for strains grouped according to their extreme habitats.

#### 2.4.1. Desiccation-Tolerant Cyanobacteria Strains B-1520 and B-1519

PCA performed for samples of two groups, where the first one was attributed to strain B-1520 and the second to the samples of all other strains studied, did not show their clear clustering (Appendix A). In contrast, PLS-DA, a statistical method that enhances intergroup variance and decreases the impact of in-group variance, demonstrated the separation of the B-1520 group from the sample population of other strains (Appendix A). This analysis also identified some metabolites in B-1520 samples that contribute to the differentiation of this strain from other cyanobacteria (Appendix A). To select metabolites of B-1520 exerting a significant difference in abundance in comparison with other strains, Volcano plot analysis was applied. This analysis indicated six metabolite features. Further, taking into consideration the different variances of compared groups and expected non-normally distributed data, the significance of the difference between groups in the abundance of metabolites indicated by Volcano analysis was additionally estimated by Welch’s *t*-test and the Mann–Whitney U-test. This approach resulted in four metabolite features (salicylic acid, erythritol, ethanolamine and an unidentified compound with RI1501) showing a marked increase in relative content in B-1520 compared to their average levels in other strains (Table 2, Appendix A). Among these compounds, salicylic acid demonstrated the most substantial fold difference (14-fold, *p* < 0.001), underscoring its potential role in the strain’s metabolic uniqueness.

PCA performed with respect to strain B-1519 did not reveal clustering with other strains. As in the previous case of B-1520, only the PLS-DA was able to detect differences important for separating strain B-1519 from other samples (Appendix A). The Volcano plot with subsequent significance validation of its results with Welch’s and Mann–Whitney tests revealed four compounds present at higher levels in this strain than in others (Table 2, Appendix A). Among these metabolites, the highest difference in abundance was found for erythritol. The level of the compound was 6.5-fold higher compared to its average content in other strains.

Interestingly, the revealed metabolite patterns for the desiccation-tolerant strains B-1519 and B-1520 shared a common propensity to accumulate two specific substances, namely erythritol and the unidentified compound RI1501.

#### 2.4.2. Strains Tolerant to High or Cold Temperatures

When comparing metabolomes of thermotolerant B-1213 against other extremophilic and extremotolerant strains, only PLS-DA, but not PCA, was effective in clustering, and therefore in finding meaningful differences in its metabolite profiles from other strains (Appendix A). Further application of a Volcano plot supplemented with additional abovementioned statistical tests allowed the identification of nine metabolites in this strain characterized by enhanced abundances compared to the average content of these metabolites in other strains (Appendix A). In this pattern, the following metabolites displayed the highest relative levels: ADP-ribose (27-fold), RI3462 trisaccharide (25-fold), unknown compound RI1022 (56-fold) (Table 3, Appendix A).

In similar comparative analyses performed for cold-tolerant strains B-1533 and B-1535, again, PLS-DA, but not PCA, was able to efficiently identify metabolite profiles that distinguished those strains from other groups. Univariate statistical analysis revealed 15 metabolites (represented by 19 features) in B-1533 and 9 metabolites in B-1535 with significantly elevated abundances compared to the average levels of those compounds observed in other strains (Table 3, Appendix A). This pattern for B-1533 was dominated by monosaccharides and their phosphate derivatives, whereas the pattern for B-1535 included an equal number of representatives of different chemical classes. The highest (at least 5-fold higher than in the group of samples from other strains) relative abundance levels were characteristic as follows: for B-1533, 2C-methyl-*D*-erythritol (17-fold), glucose (12-fold), nonadecan-1-ol (9-fold), ureidosuccinic acid (9-fold), aconitic acid (9-fold) and digalacturonic acid (5-fold); for B-1535, mannitol (10-fold), ureidosuccinic acid (10-fold), cGMP (6-fold) and MEP (5-fold). The cold-tolerant strains accumulated four common metabolites, namely 3-dehydroshikimic acid, digalacturonic acid, ureidosuccinic acid and MEP (Table 3). However, no shared metabolites were found between high- and low-temperature-tolerant strains.

#### 2.4.3. Halo(alkali)philic and Natronophilic Cyanobacteria Strains

The group of halo(alkali)philic and natronophilic strains of cyanobacteria was separated into two subgroups according to the PCA results (Figure 1). The first included strains characterized by metabolome similarities, while the second one included strains displaying significant metabolome differences. Therefore, to better reflect metabolites contributing to the observed metabolome similarities and differences in the respective subgroups, the results of the comparative analysis for the strains of these two subgroups were also considered separately.

##### Strains Demonstrating Similarity in Metabolomes

For the four strains (B-2037, B-353, B-2050 and B-1526) of the first subgroup of halo(alkali)- and natronophilic cyanobacteria, PCA performed to distinguish each specified strain did not reveal (with an exception for B-2037) essential differences in their metabolite profiles compared to other cyanobacteria strains (Appendix A). However, employing the PLS-DA method confirmed the uniqueness of each of these strains by clustering metabolomes of their samples from the sample population of other strains (Appendix A). For each of these strains, further univariate analysis revealed a pattern of specific metabolites, which mainly exhibited significantly (*t*-test *p* ≤ 0.05, U-test *p* ≤ 0.01) more than 2-fold increased abundances in the specified strains compared to the average level of the compounds in other investigated cyanobacteria (Appendix A). Thus, the pattern with the highest number of strain-distinguished metabolites within this subgroup, namely eleven, was found in strain B-2037 (Appendix A). In this pattern, ten metabolites displayed 2–10-fold accumulation in the strain. The abundance of another item (3-hydroxypiruvate) was significantly lower than the average level of the compounds in other strains. Among the accumulated metabolites, nucleotides and sugars dominated (Table 4, Appendix A). The highest abundance levels were found for glucosylglycerol (10-fold) and monounsaturated *trans*-9-hexadecenoic acid (9-fold) (Table 4, Appendix A).

The metabolite patterns associated with other strains of this halo(alkali)philic and natronophilic cyanobacteria subgroup such as B-353, B-2050 and B-1526 included eight, nine and six compounds, respectively. For strain B-353, the metabolite displaying the highest abundance was adenosine (Appendix A). The relative level of this compound was 41-fold higher its average content in other cyanobacteria (Table 4). Other metabolites accumulated in B-353 more than 5-fold were three nonannotated compounds, RI1349, RI1352 and RI1219 (Appendix A).

In the pattern of strain B-2050, the nine metabolite features that exhibited significant enrichment were isocitric acid, glycerol and representatives of sugars, nucleobases, nucleotides and their derivatives, and the unknown compound RI3201. Among these compounds, more than 5-fold accumulation was observed for digalacturonic acid (13-fold), NADPH (8.8-fold), ribonic acid (6.5-fold) and orotic acid (5-fold). The metabolic profile of strain B-1526 comprised carboxylic acids (lactic acid, 3-hydroxybutyric acid), sugars and derivatives (glucosylglycerol, RI3124 disaccharide-derivate) and pyroglutamic acid. Among them, the highest abundance in comparison with other cyanobacteria was observed for 3-hydroxybutyric acid (20-fold). Another metabolite with an 8.5-fold increase above its average level was glucosylglycerol (Table 4). Interestingly, among these four strains considered, the observed specific metabolite patterns had only a couple of common metabolites: glucosylglycerol, with elevated contents in B-2037 and B-1526, and chloride, whose relative contents were found to be the highest in B-2037 and B-353 (Table 4, Appendix A).

##### Cyanobacterial Strains Demonstrating Considerable Differences in Metabolomes

PCA and particular PLS-DA performed in regard to strain B-1529 showed clustering metabolomes of its samples from the majority of other cyanobacterial strains (Appendix A). This indicates substantial differences in metabolite profiles between these compared groups. Indeed, further investigation using Volcano analysis and statistical Welch’s and Mann–Whitney tests revealed 67 metabolites (presented by 76 features) significantly enriched in strain B-1529 compared to their average levels found in other cyanobacterial strains. The compounds were mainly representatives of amino acids, sugars, sugar phosphates, fatty acids, nucleosides and nucleosides (Table 5, Appendix A). Interestingly, about half of the compounds displayed more than 10-fold accumulation in this strain. Among them, there were 14 amino acids. The amino acids exhibiting the highest abundances (30–80-fold) were Try, Gln, The, Phe, Met.

Similar to B-1529, other two strains of this haloalkaliphilic and natronophilic subgroup, namely B-287 and B-256, also demonstrated apparent clustering from the major population of compared cyanobacterial metabolomes in corresponding PCA and PLS-DA models (Appendix A). However, metabolite patterns revealed by univariate analysis for each of the cyanobacterial strains contained a fewer strain-distinguished compounds than B-1529. Thus, for the strains B-287 and B-256, the patterns included 53 and 25 metabolites, respectively (Appendix A).

In the metabolite pattern associated with B-287, nucleotides, carboxylic acids and C2-, C3-phosphates dominated (Appendix A). In this strain, the levels of ten metabolites were over 10-fold higher compared to their average levels of other investigated strains. Among them, the most accumulated compounds were phosphoenolpyruvate (27-fold), trehalose (26-fold), a few participants of glycolysis such as 3- and 2-phosphoglycerates (14- and 18-fold) and an unidentified sugar (RI2304, 12-fold) (Table 5, Appendix A).

The metabolite pattern associated with B-256 was dominated by nucleotides and sugar phosphates (Appendix A). In this pattern, there were only three metabolites, representatives of sugar phosphate group, namely 2-deoxyribose 5-phosphate, 1-deoxyxylulose 5-phosphate and glucosamine 6-phosphate, whose abundance levels exceeded the average content of the compounds in other strains by more than 10-fold (Table 5, Appendix A).

Thus, the indicated metabolite patterns for each strain from this subgroup of haloalkaliphilic and natronophilic cyanobacteria showed substantial differences in the composition and abundance levels of individual metabolites. This may point to specific metabolic strategies that these strains specifically adapted to survive under extreme conditions of high salinity and alkalinity. Nevertheless, these strains had also some degree of overlap in their metabolite patterns. For example, patterns of strains B-287 and B-256 had twelve shared metabolites, mainly represented by sugar phosphates, nucleotides and carboxylic acids (Appendix A). Other pairs of strains such as B-287 and B-1529 had five common metabolites among which nucleotides dominated, while strains B-256 and B-1529 shared just one metabolite, dTMP. Importantly, no metabolites were found to be common to the patterns of the all three strains (B-1529, B-256, B-287), indicating a substantial difference in the metabolite patterns of the studied strains of this subgroup.

Overall, among all twelve strains considered here, B-1529 was characterized by the richest pattern of metabolites, the majority of which were representatives of amino acids. This abundance of amino acids is probably accounted for by its nitrogen-fixing ability. Notably, B-1529 is the sole diazotrophic heterocystous strain that represents the order *Nostocales* among the haloalkaliphiles and natronophiles. Here, it is important to remember that other nitrogen-fixators among the strains considered here, belonging to the desiccation- and extreme-temperature-tolerant ecological groups, were also heterocystous representatives of the order *Nostocales*. Furthermore, out of the all heterocystous diazotrophs, strain B-1529 demonstrated the most intensive accumulation (>10-fold) of rich patterns of amino acids and other many metabolites. Thus, strain B-1529 exhibits uniqueness in the diversity of enriched metabolites and in the magnitude of their enrichment.

In contrast, among the strains considered here, B-1520 from the desiccation-tolerant group (exhibiting the same heterocyst-forming capacity as B-1529) was characterized by the scarcest strain-specific pattern, containing only four metabolites.

### 2.5. Identifying Primary Metabolomic Characteristics Associated with Heterocyst-Forming Cyanobacterial Strains

As the PCA separated the six diazotrophic heterocystous strains (B-1520, B-1519, B-1213, B-1533, B-1535, B-1529) from the other strains examined in this study (Figure 1), and these nitrogen-fixing cyanobacteria appeared to be very distinct in terms of the numbers of strain-specific metabolites, this prompted us to ascertain the metabolomic signatures that differentiate the heterocystous strains and strains unable to form heterocysts. Towards this end, the primary metabolomes of these two groups of strains were compared using Volcano analysis with predefined thresholds of FC ≥ 2 and Student’s *t*-test *p*-value (FDR-adjusted) ≤ 0.05. This comparison revealed a rich pattern of metabolites, among which 23 and 41 compounds displayed higher and lower abundances, respectively, in the heterocyst-forming strains (Table 6, Appendix A). It is noteworthy that sugars, polyols and sugar phosphates, but not nitrogen-containing compounds, were predominant among metabolites displaying enhanced abundance in the heterocystous diazotrophs. The nitrogen-containing group of metabolites was represented by only four compounds, including ethanolamine, tyrosine, argininosuccinate and ureidosuccinate. Notably, the levels of the last two of these metabolites increased considerably (28- and 37-fold) in the heterocyst-forming strains (Table 6). In addition, the presence of a variety of nucleotides among compounds displaying decreased abundances was unexpected. The relative contents of the nucleotides were 3–17-fold lower in heterocystous than in non-heterocystous strains.

Other metabolites whose levels were found to be substantially elevated in the nitrogen-fixing heterocyst-forming cyanobacteria included salicylic acid (18-fold) and metabolites of isoprenoid backbone (MEP) biosynthesis such as 2C-methyl-*D*-erythritol (43-fold) and 2C-methylerythritol 4-phosphate (31-fold). Conversely, the levels of two other MEP biosynthesis downstream metabolites, namely HMBDP and isopentenyl pyrophosphate, were 8.9- and 10-fold lower, respectively, in these strains. Furthermore, the levels of C2-, C3-carboxylic acid phosphates, sugar biphosphates, trehalose, and glucosylglycerol were found to be more than fivefold lower in heterocystous than in non-heterocystous strains (Table 6).

### 2.6. Pathway Analysis

In view of the considerable range of metabolites accumulated by the haloalkaliphilic strains B-1529, B-287 and B-256, it is important to identify the molecular mechanisms that are most significant in maintaining the life processes of these cyanobacterial strains and their dependence on high-Na^+^-concentration and high-pH conditions. For this purpose, the identified primary metabolites from observed strain-specific metabolite patterns were merged into a matrix and underwent pathway analysis. This analysis combines pathway enrichment (global test) and pathway topology (relative centrality test) analyses to highlight the pathways via which the selected metabolites might support the life of haloalkaliphilic cyanobacteria strains. The pathway analysis was based on the *Synechococcus elongatus* PCC7942 pathway library available from the KEGG web resource. It is important to note that several identified substances, which are characteristic of B-1529 (glucosylglycerol and methylmalonyl-CoA) and of B-287 (*D*-erythronic acid), were not included in the results of the metabolic pathway analysis. The reason is that these substances are not available or have not been identified in the KEGG database. A further discussion of their role in cyanobacterial metabolism will be based on an analysis of relevant literature.

Many significant (*p* ≤ 0.05) pathways were identified for each of the haloalkaliphilic strains (B-1529, B-287 and B-256) (Appendix A). Among them, two pathways with the highest PI values as 1 were found to be common in at least two of the three strains: galactose metabolism for strains B-1529 and B-256, inositol phosphate metabolism for strains B-287 and B-256. This information on common important metabolomic pathways found points to possible common adaptations that evolved in similar extreme habitats. However, the most important pathways as indicated by the highest values of PI = 1 and −log10(*p*) > 5 were found to be unique in the indicated strains as follows: alanine, aspartate and glutamate metabolism in B-1529; butanoate metabolism in B-287; and pentose and glucuronate interconversions in B-256 (Appendix A). Each metabolic pathway determined to be highly reliable is associated with a group of crucial metabolites that play a significant role in these pathways (Appendix A).

## 3. Discussion

Twelve cyanobacterial strains investigated in this work were isolated from ecologically various extreme environments. These cyanobacteria are extremophilic or extremotolerant depending on the location of their optimal growth range and how wide it is in relation to environmental parameters (temperature, water availability, salinity and alkalinity) [4]. Namely, out of the twelve strains, six were haloalkaliphilic (*Sodalinema* sps (B-2037, B-353), *Limnospira* sps (B-1526, B-287, B-256), *Nodularia* sp. B-1529) being isolated from salt alkaline and soda lakes and one was halophilic as being a marine strain (*Sodalinema stalii,* B-2050). Having evolved under conditions of high salinity, these organisms formed salt tolerance which made them salt-dependent [43]. Indeed, these strains cannot survive without high Na^+^ concentrations as their proteins lose stability [44]. The marine strain originating from coastal shoals, an intertidal environment with moderately fluctuating salinity, is capable of tolerating this range of salinities. In addition, this strain, like all marine strains, requires not only Na, but also C1^−^, Mg^2+^ and Ca^2+^ [39]. Furthermore, salt alkaline and soda lakes are characterized by not only various salinity fluctuations, but also high concentrations of HCO_3_ and CO_3_^2−^ and pH 9–10 [31]. Therefore, the haloalkaliphilic strains are restricted to narrow ranges of alkalinity fluctuations. Given these considerations, the cultivation media for the halo(alkali)philic strains were selected to provide optimal conditions for photoautotrophic growth. However, to maintain long-term storage, the cyanobacteria were kept in low light conditions and without additional CO_2_ supply. Thus, this group of cyanobacteria presents Na^+^- and alkalinity (except marine strain *S. stalii*)-requiring strains of various adaptational types.

Five other cyanobacterial strains considered as extremotolerant are representatives of terrestrial macrocolony *Nostoc commune* (B-1520, B-1519), able to sustain desiccation, thermotolerant *Dolichospermum* sp. B-1213 from a hot spring environment, and cold-tolerant cyanobacteria *Anabaena* sp. B-1535 and *Anabaena* cf. pirinica B-1533, able to sustain low positive temperature and high-rate water flows. These soil and freshwater cyanobacteria strains can survive over a wide range of fluctuations in one or more environmental factors. However, they are unable to develop under such wide factor fluctuations. For the majority of the strains, optimal conditions for growth are not extremely harsh. Nevertheless, their high adaptability potential allows them to withstand long periods of desiccation, heat and cold. This remarkable adaptability has sparked scientific interest in utilizing these strains in biotechnological applications [45,46]. Similar to halo(alkali)philic strains, the extremotolerant cyanobacteria were maintained under reduced light and atmospheric CO_2_ to ensure long-term storage conditions. It is noteworthy that all the cyanobacterial strains were unialgal but non-axenic as they had been co-isolated with other bacteria from the same habitat. Despite the fact that cyanobacteria constituted the dominant component in each culture, their primary metabolome that was studied could also be influenced by the bacterial cohabitants.

Further experimental design was thoughtfully considered to address the following primary questions of this exploration work: Are there constitutive metabolic differences between strains in the absence of the pressure of environmental specialization factors under culture storage conditions? If such differences are identified, could they be related to the extreme conditions of origin of the strains? What is the pattern of strain-specific metabolites that might indicate the strains’ extreme origin after their prolonged storage?

The responses to the first two posed queries were derived through an examination of cyanobacterial GC-MS- and LC-MS-based metabolomic data, employing multivariate statistics and hierarchical clustering (Figure 1 and Figure 2). All strains studied here exhibited distinct clustering in the PCA and heatmap models, indicating the presence of strain-specific metabolomic characteristics. These latter reflect constitutive (inherited) genetic differences between the strains. Their occurrence can be explained by the position of ‘plasticity-led evolution’. This suggests that environmentally induced alterations in an organism’s phenotype may facilitate the emergence of new adaptations if they are consistently advantageous to the organism in the new environment. In this case, selection may lead to an evolutionary loss of plasticity and, consequently, to a reduced sensitivity to the original environmental trigger. Finally, this could culminate in the constitutive expression of the trait that was previously induced by the environment [47].

Furthermore, PCA results reflected inherent similarities in the metabolomes of isolates from similar environments and general differences in the metabolomes of representatives of different ecological groups. Indeed, the similarities in metabolomes found in organisms that evolved under similar environments can reflect the evolution of their common acclimation strategies which were traced to changes in their genome [11,15,48]. The genetic basis of cyanobacterial adaptation to specific habitats has been investigated in a comparative genomic analysis of 650 cyanobacterial genomes from various ecosystems [48]. The general strategy of marine strains was to shrink genomes adapting to nutrient-poor conditions. Terrestrial strains expanded genomes partly due to horizontal gene transfer to cope with environmental fluctuations.

In addition, the present work has yielded several noteworthy outcomes concerning halo(alkali)philic and natronophilic strains. Firstly, as would be expected, some phylogenetically close strains, e.g., *Sodalinema* strains, have been shown to share certain metabolome characteristics under long-term storage, despite their initially different ecological environments (Table 1). This is evidenced by the separation of the strains in a cluster in the PCA model (Figure 1). Thus, *Sodalinema stalii* B-2050, a strain that is only halophilic, but not alkaliphilic and natronophilic, appeared to be in the same cluster as other haloalkaliphilic and natronophilic *Sodalinema* strains (Figure 1, Cluster 3.1, Appendix A). In contrast, other closely related strains, such as *Limnospira* strains (Appendix A), which were isolated from ecologically similar but geographically disparate environments (Table 1), have been observed to be distributed by PCA across different metabolomic clusters (Figure 1). One of the strains (B-1526) was located in close proximity to the *Sodalinema* strains (Figure 1, Cluster 3.1). Two other *Limnospira* strains (B-256 and B-287) along with a heterocystous *Nodularia* strain formed a separate group (Figure 1, Cluster 3.2) that differed from the others by considerable intergroup metabolome variability. This group of strains (B-287, B-256, and B-1529) was distinguished by the highest number of various accumulated metabolites, as demonstrated by the heatmap (Figure 2). Consequently, the strains of the genus *Sodalinema* exhibit minimal variability in their metabolomes, while strains of the genus *Limnospira*, despite their close evolutionary relationships, display considerable variability in their metabolomes. This finding might be explained by at least one of the following reasons: (1) A heightened sensitivity of *Limnospira* strains to the influence of diverse extreme habitat conditions on the formation of strain-specific adaptations, as compared to *Sodalinema* strains; (2) more variable natural environments in the case of *Limnospira* than *Sodalinema* strains, which can reflect the evolution of more various acclimation strategies in the former; (3) different capacities of *Limnospira* and *Sodalinema* strains to withstand long-term storage cultivation.

It is also important to note that PCA enabled the identification of sample metabolomes representing diazotrophic heterocystous strains from the entire metabolomic dataset (Figure 1). All six diazotrophic heterocystous strains examined in this study (B-1520, B-1519, B-1213, B-1533, B-1535, and B-1529) are classified within the order *Nostocales* and are distinguished by their capacity to form heterocysts for effective atmospheric nitrogen fixation [49]. Remarkably, these strains, despite being isolated from ecologically disparate extreme environments, exhibited a certain degree of similarity in their metabolomes, as revealed by PCA. This metabolomic similarity may be attributable to their shared capacity for heterocyst formation for nitrogen fixation. It is noteworthy that, in this case, this similarity between the metabolomes of distantly related strains is indicative of a shared, ancient genetic heritage rather than the evolution of analogous strategies for adaptation to extreme environmental conditions. This functionality (heterocyst-forming ability) is likely to be evolutionarily advantageous due to its persistence in cyanobacteria under diverse extreme environmental conditions. Therefore, it can be hypothesized that the distinguishing pattern of metabolites that were revealed between the metabolomes of heterocystous and non-heterocystous strains is indicative of alterations in the primary metabolism of cyanobacteria associated with heterocyst formation and N_2_ fixation. Interestingly, in this pattern of metabolites, among the nitrogen-containing metabolites, a variety of nucleoside di- and triphosphates exhibited notable abundance decreases, whilst only a small number of amino acids (in particular, argininosuccinate and ureidosuccinate) displayed significant increases in heterocystous cyanobacteria (Table 6). The former observation suggests a high level of energy consumption in heterocystous strains, as maintaining heterocysts (which ensures N_2_ fixation and subsequent distribution of the fixed nitrogen in cell metabolism) requires a high energy investment [50,51]. The two discovered metabolites, whose contents were many times higher in heterocystous strains, may indicate two mutually active pathways involved in the distribution of fixed nitrogen within the metabolism of these cyanobacteria. Specifically, the argininosuccinate, a precursor of arginine (a nitrogen-rich amino acid), suggests the involvement of the arginine–ammonia cycle [51], which, in combination with aspartate, may further contribute to the formation of a cyanophycin co-polymer, serving as a nitrogen reservoir. Conversely, the enhanced abundance of argininosuccinate in heterocystous cyanobacteria may also be attributable to the inhibition of arginine biosynthesis by a feedback mechanism [52] under strain storage conditions. The enhanced abundance of ureidosuccinate (also known as carbomoyl-*L*-aspartate) indicates the transfer of fixed nitrogen in the pyridine biosynthetic pathway. Another interesting outcome of the comparison is that heterocystous cyanobacteria exhibit elevated levels of sugar monophosphates, while non-heterocystous cyanobacteria display higher levels of sugar diphosphates. This observation may be indicative of the differential contribution of the pentose phosphate pathway and the Calvin cycle in the metabolism of the respective cyanobacterial groups.

To identify the pattern of strain-specifically accumulated metabolites, we compared the metabolome of each strain with the average levels of metabolic characteristics of the other strains studied. Interestingly, few (up to four) strain-specific metabolites were identified in desiccation-tolerant terrestrial strains (Appendix A). Their numbers were higher (8–15) in hot- and cold-extremotolerant strains (Appendix A) and varied considerably (6–67) among halo(alkali)philic strains (Appendix A). This difference in the number of constitutively up-regulated metabolites between extremotolerant and extremophilic strains may reflect differences in their acclimation strategies. Indeed, terrestrial and freshwater extremotolerant strains isolated from turbulent ecosystems involve, on average, more inducible genes of more functional categories in shaping adaptation to environmental fluctuations [48]. Therefore, in the absence of the influence of such environmental fluctuations, no end products (metabolites) of these genes are formed in the strains under storage. In this context finding the patterns of strain-specifically accumulated metabolites indicates some originally environment-inducible biosynthetic mechanisms could lose sensitivity to the fluctuating environmental factor and, finally, result in the constitutive expression of the metabolic traits. On the other hand, organisms (extremophiles) adapted to thrive in less variable environments have a greater number of constitutively expressed genes enabling mechanisms of evolved acclimatization [53]. Their final metabolite products are expected to be synthesized in the strains during their storage cultivation.

### 3.1. Constitutive Patterns of Metabolites and Their Adaptive Potential for Extremotolerant Cyanobacteria

*Nostoc commune* strains B-1520 and B-1519 were isolated from terrestrial macrocolonies subjected to desiccation in natural habitat regularly and are tolerant to drying–rehydration cycles. Although desiccated cyanobacteria are incapable of photosynthesis, rehydration rapidly restores (in just 5 min!) their photosynthetic activity [54]. Extracellular polysaccharides (EPSs) produced in response to stress play a crucial role in their tolerance mechanisms [54,55,56,57].

An important metabolite found in significant amounts in B-1520, but not in B-1519, was salicylic acid (Table 2). This key signaling metabolite is known to activate defense responses in plants [58,59,60,61]. The accumulation of salicylic acid in the *N. commune* strain B-1520 agrees with the data of Toribio et al. [62]. They indicated the production of salicylic acid by 28 cyanobacteria strains and demonstrated that some strains of the genus *Nostoc* exhibited the most significant level of salicylic acid production. It is possible that salicylic acid acts as a trigger to enhance the EPS production in B-1520 under storage conditions. This can be explained by the fact that the production of EPSs (especially capsule EPSs bound to the cell surface) in *N. flagelliforme* [63] could be enhanced by salicylic acid added to the culture medium. Salicylic acid first increased the NO level which in turn promoted EPS biosynthesis. The latter occurred via the nitrosylation of EPS biosynthetic enzymes [64]. Additionally, the cyanobacteria’s ability to synthesize enhanced levels of salicylic acid can be used in agrobiotechnologies to promote plant disease resistance [65,66]. It also cannot be excluded that salicylic acid could at least partly be produced by bacteria accompanying the strain culture. It is known that soil strains of *N. commune* may form associations with heterotrophic bacteria, of the Actinobacteria phylum [67], with some members of this group such as *Streptomyces* having been shown to produce salicylic acid [68].

Another important metabolite that was accumulated in both B-1520 and B-1519 under storage conditions is erythritol (Table 2). Erythritol was found in the composition of EPSs from *N. commune* [69,70]. In addition, some biotechnological strategies have created genetically modified cyanobacteria that produce erythritol [71]. Erythritol is produced from erythrose, previously dephosphorylated from erythrose-4-phosphate (E4P). It is noteworthy that E4P, an intermediate of the Calvin and pentose phosphate oxidation (PPO) pathways, is the common precursor for two metabolites, erythritol and salicylic acid, found in this work as constitutively up-regulated in at least one desiccation-tolerant strain. These data suggest that carbon flux in those strains might be directed via E4P to two routes. The first is the shikimate pathway producing phenolic compounds, which mainly are known as antioxidants neutralizing free radicals formed during desiccation. The second is a branch reaction producing erythritol which might serve as an EPS constituent or an osmoprotectant, protecting cellular membrane integrity and proteins from denaturation under water-deficient conditions [72].

*Dolichospermum* sp. B-1213 is an isolate from a hot spring habitat (Table 1). In this work, storage cultivation of the strain was carried out at room temperature (22 °C), whereas the optimal growth conditions are in the range of 35–38 °C [32]. Therefore, in addition to reducing the light and CO_2_ supply to limit cyanobacteria growth under storage, the temperature shifting from optimal to low values could act as an additional stress factor for B-1213 and cause specific metabolic responses of the strain cells. Indeed, the highest accumulation of ADP-ribose (27-fold) and NADH (4-fold, Table 3) may indirectly indicate the presence of this stress factor. High levels of ADP-ribose in cells are toxic because of its ability to cause nonenzymatic ADP-ribosylation inactivating proteins [73,74]. Elevated NADH levels can indicate that its oxidated state NAD^+^ (the dominant NAD form in unstressed cells) can be depleted in glycolysis. An important role of glycolysis in the B-1213 is also evidenced by the accumulation of glucose 1-phosphate, a product of glycogen degradation, which enters glycolysis via conversation to glucose 6-phosphate (G-6-P). Furthermore, the accumulation of glutamate in the strain may indicate that the carbon flux derived from glycogen consumption is redirected for amino acid synthesis. This is consistent with the literature data [75] showing that redirection of carbon flow from glycogen synthesis in *Synechococcus elongatus* led to enhanced glutamate production.

The cold-tolerant *Anabaena* sp. B-1535 and *A.* cf. pirinica B-1533 (Table 1) were maintained at 22 °C during their storage. This meant that the growth of the strains was not limited by temperature fluctuations that the strains had adapted to, but only by the light and CO_2_ supply. Under these conditions, both strains exhibited constitutive up-regulation in a range of metabolites (Table 3). Many of the metabolites can be related to EPS and glycolipid production during heterocyst formation in an environment lacking a fixed nitrogen source [38,76]. Heterocysts, N_2_-fixing cells, sustain an O_2_-limiting micro-environment for oxygen-sensitive nitrogenase by forming a dense envelope of glycolipids and EPSs [77,78]. The final step of glycolipid synthesis includes the reaction of glycolipid synthase transferring glucose to the fatty alcohol [77]. Thus, we proposed that the elevated levels of hexoses, their phosphates and fatty alcohol (nonadecan-1-ol) observed in at least one of the B-1533 and B-1535 strains may be involved in the synthesis of heterocyst glycolipids. Meanwhile, hexoses, their derivatives and digalacturonic acid may be involved in the synthesis of EPSs including heterocyst polysaccharides.

B-1533 shows more up-regulated metabolites than B-1535, indicating a more active metabolic status. This may be due to the different capacities of the strains to cope with photorespiration. The enhanced glycolate level in B-1535 suggests enhanced RUBISCO oxygenase activity in this strain.

B-1533 showed elevated levels of metabolites, members of all three principal metabolic pathways of glucose catabolism: the Embden–Meyerhoff–Parnas (EMP or glycolysis), OPP and Entner–Dudoroff (ED) pathways, as well as the TCA cycle (Table 3) [79]. These pathways may provide energy and carbon flux towards N_2_ fixation and biosynthesis of fatty acids, alcohols, EPSs, and various secondary metabolites. Indeed, the accumulation of compounds such as 2C-methyl-*D*-erythritol and 2C-methyl-*D*-erythritol-4-phosphate (MEP) in B-1533 and/or B-1535 may indicate active isoprenoid biosynthesis [80]. In addition, the potential contribution of bacteria associated with these cyanobacterial cultures to the elevated levels of these two metabolites cannot be dismissed [81]. Additionally, increased levels of 3-dehydroshikimic acid, an intermediate of the shikimate pathway, indicate active biosynthesis of aromatic amino acids and diverse phenolic metabolites. Both strains also exert high levels of ureidosuccinate, an intermediate of de novo pyrimidine biosynthesis. This indicates the direction of nitrogen flux fixed by nitrogenase from amino acids to pyrimidine nucleotides [82].

### 3.2. Constitutive Patterns of Metabolites and Their Adaptive Potential for Haloalkaliphiles and Natronophiles

The marine *S. stalii* (B-2050), the haloalkaliphiles and natronophiles *S. gerasimenkoae* (B-353) and *S. orleanskyi* (B-2037) from salt alkaline lakes, and *Limnospira* sp. (B-1526) from a soda lake habitat exhibited less pronounced metabolic differences among themselves under storage conditions compared to three other strains from soda lake environments: *Limnospira* sps B-287 and B-256 and *Nodularia* sp. (B-1529) (Figure 1). This finding enabled us to distribute all seven halo(alkali)philic strains between two groups according to the relative similarity in their metabolomes.

Despite the general metabolic similarity (Figure 1), the cyanobacteria strains from the first group exhibited distinctive metabolite levels showing the strains’ unique adaptations to specific habitats. B-2037 demonstrated the accumulation of glucosylglycerol (Table 4) which acts as compatible solute; i.e., by increasing cellular osmolality, it contributes to the osmotic acclimatization of cells [53]. This compound also protects membrane stability and proteins from denaturation under high salinity and alkalinity. In addition, elevated related abundance of phosphate and ADP in B-2037 can at least partially be related to glucosylglycerol biosynthesis [83]. Three other metabolites from the pattern of B-2037-distinguished metabolites, up-regulated sedoheptulose-1,7-biphosphate and 2-phosphoglycolate and down-regulated 3-hydroxypyruvate, indicate activities of the Calvin cycle and photorespiration. The latter pathway may be stimulated by the limited light supply to maintain the strain storage conditions. To support CO_2_ fixation by RUBISCO (and therefore to decrease photorespiration) under high-alkalinity conditions and to utilize the available HCO_3_^−^, haloalkaliphilic cyanobacteria have evolved carbon concentration mechanisms [84]. These mechanisms enable the active transport of HCO_3_^−^ into the cell where it is converted to CO_2_ in specialized protein carboxysomes containing RUBISCO [85]. The elevated ADP levels observed in this strain may indicate energy consumption associated with the described process. The elevated Cl^−^ level in B-2037 may have toxic effects on cellular metabolism, potentially due to its effects on protein structure [53]. The *trans*-isomer of 9-hexadecenoic acid, which was identified as accumulating in the strain, may have been produced by bacteria associated with the culture of the strain rather than by the strain itself. This is because *S. orleanskyi*, like other cyanobacterial strains, is characterized by a high content of only the *cis* but not the *trans* isomer of 9-hexadecenoic acid [86].

For B-353, the elevated level of shikimate may be related to the carbon flux directed towards the biosynthesis of aromatic secondary metabolites [87]. Additionally, this strain exhibited the accumulation of adenosine (41-fold higher than the mean level of this metabolite across all strains studied, Table 4). This was unexpected, and further investigation is needed into the role of this metabolite in tolerance mechanisms. However, this result is in line with other studies also indicating that cyanobacteria are capable of producing enhanced level of adenosine [88,89]. Fatima et al. showed a genetically modeled strain of *Synechococcus elongatus* which responded to salt stress by the accumulation of various nucleosides, including adenosine [88]. Moreover, adenosine produced by cyanobacteria has promising biotechnological potential, as it exhibits cytotoxicity and is used as an anti-leukemic agent [89,90].

In the marine strain B-2050, up-regulation of glycerol, galacturonic acid and several other sugar derivatives may be associated with EPS biosynthesis, which according to some data may be associated with increased NaCl tolerance [91]. B-2050 also exhibited the highest levels of orotic acid and NADPH among the other strains studied. A high level of orotic acid may reflect constitutive activity of de novo pyrimidine synthesis. Elevated levels of NADPH may indicate low activity of the Calvin cycle, in the reactions of which NADPH can be utilized.

The soda lake strain B-1526 has demonstrated the accumulation of a considerable quantity of glucosylglycerol, functioning as a compatible solute, similar to that observed in B-2037. B-1526 also displayed the highest level of 3-hydroxybutyric acid (Table 4), indicating that it is the most effective producer of poly-3-hydroxybutyrate (PHB) among the strains examined in this study. PHB is a non-water-soluble energy-rich polymer that serves as a specific carbon and energy reserve for withstanding nitrogen starvation under stress conditions [92,93].

The second group, comprising the three haloalkaliphiles B-1529, B-287 and B-256 derived from soda lake habitats, displayed the most distinctive metabolomes (Figure 1) and the richest patterns of diverse accumulated metabolites (Figure 2, Appendix A). The *Nodularia* sp. B-1529, which is the only diazotrophic heterocystous strain among other haloalkaliphiles studied here, exhibited the highest number of accumulated metabolites in its primary metabolome (Table 5). The metabolites were participants of the major catabolic (EMD, ED, OPP, TCA), biosynthetic (e.g., de novo pyrimidine and purine nucleotide, shikimate, EPS, MEP, fatty acid pathways) and remodeling routes (fatty acid desaturation). Additionally, hexoses, sucrose and glucosylglycerol are compatible solutes ensuring high cellular osmolality and homeostasis under high-Na^+^-concentration and high-pH conditions. It is noteworthy that in B-1529, the levels of the majority of amino acids were higher than those in the other heterocystous diazotrophs studied in this work (Appendix A). It might be speculated that amino acids such as arginine and aspartate may be involved in the metabolism of cyanophycin, serving as a nitrogen reserve in cyanobacteria [94]. On the whole, the elevated levels of many nitrogen-containing metabolites (amino acids, nucleosides, nucleotides) point to a high cellular nitrogen status [95,96]. This suggests that the N_2_-fixation process is highly efficient in B-1529. It is notable that *Nodularia* is notorious for its production of toxic non-ribosomal pentapeptides, nodularins [95,97,98,99]. These toxic compounds may act as liver tumor promotors [100]. The amino acids produced by this strain might be used for nodularin biosynthesis. The production of the nodularin is not constitutive, but rather inducible [95]. However, the conditions that influence its biosynthesis as well as the biological role of the compounds have not been well elucidated [101,102]. As nodularin is a nitrogen-rich compound, its biosynthesis depends on nitrogen availability and also requires active photosynthesis [103]. Nodularin binds to proteins. The functionality of the binding has been discussed in the context of ROS protective stabilization of proteins by nodularins or as a means for nitrogen storage [95].

Moreover, pathway analysis was employed to elucidate the most efficient pathways among those discussed herein. This enabled the identification of the most efficient pathway in the strain under storage conditions, based on its −log_10_(*p*) and PI values, as alanine, aspartate and glutamate metabolism (Appendix A). This result was anticipated, as this pathway reflected the aforementioned metabolism of amino acids resulting from nitrogen fixation and their potential utilization for nodularin biosynthesis.

The genus *Limnospira* to which the two strains B-256 and B-287 (and also previously described B-1526) belong, in contrast to *Nodularia*, is not a toxic cyanobacteria genus [104]. *Limnospira*, also designated as Spirulina, is acknowledged as a rich source of diverse nutrients [105]. It usually exhibits optimal growth and a high quality of other nutrient parameters when cultivated under 30 ± 5 °C [106]. Indeed, these facts are consistent with our observations, which revealed a marked abundance of various important primary metabolites in both strains even under storage conditions (Appendix A).

Strain B-287 is characterized by a high energetic status, as can be concluded from its highest relative content of ATP and GTP (Table 5, Appendix A). The high ATP and 2- and 3-phosphoglycerate levels may indicate that carbon flow is directed not to energy production but rather towards biosynthetic processes. This is evidenced by the relatively high content of glycerol 3-phosphate, acetyl-CoA and malonyl-CoA in B-287, which may be involved in polyketide synthesis [107,108], fatty acid metabolism, and PHB production [109]. Furthermore, pathway analysis confirmed the importance of these metabolic pathways. It identified ‘butanoate metabolism’ as the most crucial for B-287 cultivated under storage conditions (Appendix A). The butanoate metabolism encompasses processes involving short-chain fatty acids, thereby establishing a connection between the fatty acid, polyketide and PHB metabolisms. Additionally, B-287 exhibited elevated levels of important intermediates involved in isoprenoid biosynthesis (Table 5, Appendix A) [110]. B-287 also has the highest trehalose accumulation among the strains studied. Trehalose and sucrose, along with glucosylglycerol, provide cellular osmoprotection as compatible solutes [111]. Trehalose synthesis is supported by elevated levels of UDP- and ADP-glucose (Table 5), starts from glycogen and is induced by high salt concentrations [53].

B-256 accumulated deoxy-pentoses and deoxyribonucleotides, compounds (adenylosuccinic acid, xanthosine-5′-phosphate, inosinic acid) involved in de novo purine nucleotide synthesis and metabolites (UDP-*N*-acetylglucosamine and glucosamine 6-phosphate) associated with cyanobacterial cell wall biogenesis (Appendix A). Pathway analysis identified ‘pentose and glucuronate interconversions’ as the most significant metabolic route in this strain under storage conditions (Appendix A). This indicates the importance of sugar-phosphate substrates in nucleotide synthesis, cell wall biosynthesis and, overall, the active growth of this strain.

An overview of the aforementioned metabolic processes that may occur in extremotolerant and extremophilic cyanobacteria to maintain their viability under storage is given in Table 7.

## 4. Materials and Methods

### 4.1. Reagents

The reagents were obtained from various manufacturers as follows: *L*-aspartic acid and 2-oxoglutaric acid from Reanal (Budapest, Hungary); hexane (puriss p.a.) from Conlac GmbH (Leipzig, Germany); *N*-methyl-*N*-(trimethylsilyl)trifluoroacetamide (MSTFA, MS grade) from Macherey-Nagel GmbH and Co KG (Düren, Germany); methanol (LC grade) from Vekton (Saint-Petersburg, Russia). Other chemicals were sourced from Sigma-Aldrich Chemie GmbH (Taufkirchen, Germany). Water purification was performed using a Millipore Milli-Q Gradient A10 system (resistance 18 mΩ/cm, Merck Millipore, Darmstadt, Germany).

### 4.2. Cyanobacterial Strains’ Characterization and Cultivation

Twelve non-axenic cyanobacterial strains were obtained from the Collection of Microalgae and Cyanobacteria IPPAS (K.A. Timiryazev Institute of Plant Physiology RAS, Moscow, Russia) (Table 1). For the experiments, the strains were grown in 350 mL Erlenmeyer flasks containing 150–200 mL of appropriate liquid medium under conditions (reduced light and an absence of additional CO_2_ supply) which are used in the IPPAS collection for long-term maintenance of the studied strains (Table 1).

For the analysis, cyanobacterial biomass was collected in 50 mL tubes by centrifugation (4500× *g*, 15 °C, 10 min), the supernatant was thoroughly removed and the obtained pellets were washed twice with deionized water, frozen at −80 °C and freeze-dried in a Labconco FreeZone 6 L lyophilizer (Labconco, Kansas City, MO, USA) at a condenser temperature of −40 °C and a pressure of 100 Pa for 2–3 days.

### 4.3. GC-MS-Based Analysis of Thermally Stable Polar Metabolites

Sample preparation procedure (metabolite extraction and derivatization) and GC-MS analysis were performed according to the protocol described by Bilova et al. [112,113] with minor changes. In detail, to extract thermally stable metabolites, lyophilizates of cyanobacteria were processed in a two-step procedure. First, cyanobacteria material (approximately 10 mg) was supplemented with 800 µL ice-cold methanol containing the internal standard (IS) adonitol at a final concentration of 50 µmol/L. After intensive vortexing (30 s) and subsequent centrifugation of the resulting suspension (12,000× *g*, 4 °C, 10 min), a portion (500 µL) of the obtained supernatant was transferred to a new tube. Water (400 µL) was added to the remaining supernatant, and cyanobacteria residues were resuspended by vortexing to obtain suspension. After repeated centrifugation, the obtained supernatant was collected and combined with the first extract portion. The total volume (1100 µL) of the combined aqueous (aq.) methanolic (1:3, *v*/*v*) extract was mixed with 300 µL hexane to purify the extract from low-polarity and high-molecular-weight compounds such as chlorophylls, carotenoids and lipids which are not suitable for GC-MS. The resulting mixture was vortexed and briefly centrifugated (2 min, 5000× *g*) to separate the nonpolar hexane phase from the aq. methanolic phase containing polar metabolites. Aliquots (50 µL) from the sample aq. methanolic extract were taken for further analysis. The indicated volume of sample aliquots was selected as the optimal one in a series of preliminary experiments directed to optimize the performance of subsequent derivatization and GC-MS analysis.

The aliquots were evaporated to dryness at +4 °C using a Labconco CentriVap vacuum concentrator (Labconco Corporation, Kansas City, MO, USA). The resulting dry residues were derivatized through a two-step procedure by subsequently using methoxyamine hydrochloride (MEOX) and *N*-methyl-*N*-(trimethylsilyl)trifluoroacetamide (MSTFA), following a previously established protocol [114]. The obtained mixtures of thermally stable metabolite derivatives were analyzed by gas chromatography–mass spectrometry (GC-MS). For that 1 μL from each sample was injected with a CTC GC PAL liquid injector (Shimadzu Deutschland GmbH, Duisburg, Germany) in a GC2010 gas chromatograph. The separation of analytes was carried out in the mode of programmable temperature gradient from 40 to 320 °C on a ZB-5MS nonpolar capillary column (30 m, 0.25 mm, 0.25 μm, Phenomenex Inc., Torrance, CA, USA) coupled to a Shimadzu GCMS QP2010 quadrupole mass-selective detector equipped with electron ionization ion source and operating at the settings summarized in Appendix A.

A sample sequence (batch) organized for the GC-MS analysis involved various types of samples arranged in the following order: (i) hexane, (ii) a mixture of C_10_–C_40_ alkanes dissolved in hexane, (iii) MSTFA, (iv) a derivatization blank (containing only derivatization agents), (v) an extraction blank (containing only IS and derivatization agents), (vi) experimental samples in a randomized sequence, (vii) quality controls (QCs) obtained by mixing equal volumes (200 μL) of all analyzed samples and prepared from aliquots (50 μL) of the resulted pooled extracts and (viii) 28 mixes of 5–6 authentic standards (a final concentration of each component 50 µmol/L) serving for reliable metabolite identification. QCs were inserted into the sample batch in every 5–6 experimental samples to evaluate method performance.

The processing of the obtained GC-MS data (chromatograms) employed an untargeted approach aimed at unbiased annotation of all detected total ion current (TIC) chromatographic peaks with a signal-to-noise ratio (S/N) ≥ 3. For that, the following pieces of software were used: the Automated Mass Spectral Deconvolution and Identification System (AMDIS, version 2.66 from 8 August 2008, www.amdis.net, accessed on 25 September 2023); Xcalibur, version 3.0.63 from 5 August 2013 (TermoFisher Scientific Inc., Bremen, Germany); and MS-DIAL, version 4.9 from 1 January 2022 (RIKEN Center for Sustainable Resource Science, Kanagawa, Japan). The quality of the obtained GCMS data was evaluated by the baseline position, retention time, shape and height of chromatographic peaks and the level of noise background. The annotation of TIC peaks to specific analytes (also called features) relied on a search of experimentally obtained retention indices (RIs) and electron ionization mass spectra (EI-MS) against established reference mass spectral libraries. For each analyte, RI was calculated by retention time (t_R_) and retention indices of linear alkanes C_10_–C_40_ (Appendix A) according to the following formula indicated in the AMDIS manual instruction: RI_comp_ = RI_b_ + ((RI_a_ − RI_b_) × (t_Rcomp_ − t_Rb_)/(t_Ra_ − t_Rb_)), where t_Rb_ and t_Ra_ are the retention times of alkanes which are the closest before and after the actual retention time (t_Rcomp_) of the target compound, and RI_b_ and RI_a_ represent the retention indices associated with t_Rb_ and t_Ra_, respectively. The following libraries were used for the metabolite structural annotation: National Institute of Standards and Technology (NIST), Golm Metabolome Database (GMD) and an in-house authentic standard library (IhASL) (Appendix A). The identification of analytes was achieved by matching the obtained GC-MS data (t_R_, RI and EI-MS) with those of authentic standards from IhASL coeluted with the experimental samples. The quantitation of analyte relative abundances was conducted by integrating the corresponding extracted ion chromatograms (XICs) built for the analyte characteristic fragment ions (*m*/*z* ± 0.5) at the analyte specific retention times.

### 4.4. LC-MS Analysis of Thermally Labile Polar Metabolites

Analysis of thermally labile anionic polar metabolites was performed based on reversed-phase ion pair ultrahigh-performance liquid chromatography and tandem mass spectrometry accomplished with electrospray ionization and a triple quadrupole mass analyzer (RP-IP-UHPLC-ESI-QqQ-MS/MS), following a method described by Balcke et al. [115] with minor adaptations.

The lyophilized material of cyanobacteria was weighed, and 25 ± 3 mg DW was added to polypropylene 2 mL microtubes containing 200 mg glass beads (0.75–1 mm diameter), 3 stainless steel beads (3 mm diameter) and 1 stainless steel bead (5 mm diameter). The cyanobacteria material was extracted through a two-step procedure. The first extraction was performed with 900 µL of an ice-cold (−80 °C) dichloromethane/ethanol mixture (2:1, *v*/*v*) and 100 µL of ice-cold HCl/water (1:200, *v*/*v*). The mixture was intensively homogenized (5.0 m/s, 3 × 20 s, FastPrep-24^TM^, MP Biomedicals, Eschwege, Germany). After subsequent centrifugation (4 °C, 10,000× *g*, 5 min), the polar supernatant fraction (300 µL) was collected into a new 1.5 mL microtube. During the second extraction step, the plant material residue was resuspended in an additional portion (50 µL) of ice-cold HCl/water (1:200, *v*/*v*). After the sample mixing and centrifugation, as described above, the obtained polar supernatant fraction (120 µL) was combined with the first polar extract portion. The combined extract was evaporated to dryness at +4 °C using a Labconco CentriVap vacuum concentrator (Labconco Corporation, Kansas City, MO, USA) and stored at −20 °C. For LCMS analysis, the dried extract residue was solubilized in 180 µL of a water–ethanol mixture (1:3, *v*/*v*), and after additional centrifugation (12,000× *g*, 4 °C, 5 min) it was transferred into a glass insert of a chromatographic vial, capped with membrane-secured plastic septa. Aliquots (5 µL) of the extracts were analyzed by a Waters ACQUITY H-Class UPLC System (Waters GmbH, Eschborn, Germany), online coupled to AB Sciex QTRAP 6500 MS/MS System (AB Sciex, Darmstadt, Germany) under the chromatographic and mass spectrometric settings presented in the Appendix A (Appendix A). A sequence for the LC-MS analysis included samples of different types (experimental samples in randomized order, QC samples, samples of a multicomponent standard solution, extraction blanks). Targeted acquisition of LC-MS data was performed in the multiple reaction monitoring (MRM) mode and relied on MRM ion pairs (*m*/*z* of molecular ion and *m*/*z* of characteristic fragment ion) of 183 individual authentic standards (target compounds). For the quantitative analysis of the target compounds in the experimental samples, the extracted MRM peaks were integrated using MultiQuan^TM^ (version 3.0.2) tool (AB Sciex, Darmstadt, Germany).

### 4.5. Statistical Analysis

The integrated areas of all analyte peaks recorded by GC-MS and LC-MS methods were combined into a digital matrix. The dataset after normalization to the dry weight of samples was filtrated to remote analytes with a high percentage of missing data (i.e., analytes that were absent in ≥20% of the samples) and those exhibiting significant in-group variation (relative standard deviation (RSD) ≥ 70%). The resulting dataset was statistically processed using the online platform MetaboAnalyst 5.0 (www.metaboanalyst.ca, accessed on 1 December 2023). The methods of multivariate statistics such as principal component analysis (PCA) and partial least squares discriminant analysis (PLS-DA) were applied. Additionally, hierarchical clustering with heatmap representations and a Volcano plot were used to visualize differences in relative abundances of annotated analytes across all experimental groups and to highlight analytes displaying specific behavior (accumulation or sharp decrease in abundance and disappearance) for each analyzed cyanobacteria strain, respectively.

Univariate statistics (Volcano plot) was applied to compare the abundances of individual metabolites in each of the twelve strains studied here with a pool of samples compiled from eleven other strains. The comparison results were presented as the fold difference (FC) in the mean abundances of individual metabolites and the significance of the differences (*p*-value) between compared groups estimated by Student’s *t*-test. For FC and *t*-test *p* presented in tables and figures, the threshold values were as follows: the minimal value of FC was 2 and the maximal *p*-value was 0.05 as typically recommended for metabolomics studies [116,117]. The reliability of the observed FCs was assessed by adjusting the *p*-value with the false discovery rate (FDR) estimated with the Benjamini–Hochberg method [42]. As the variance of the groups compared was different and the data may not be normally distributed, the significance of the observed differences in the abundances of metabolites detected by the Volcano analysis was additionally assessed using Welch’s *t*-test and the Mann–Whitney U-test [116,117]. The threshold *p*-values for the tests reported in the tables were as follows: the maximum *p*-value was 0.05 for the Welch’s *t*-test and 0.01 for the Mann–Whitney U-test.

### 4.6. Metabolomic Pathway Analysis

Pathway analysis, available via the web resource Metaboanalyst 5.0, was used to evaluate the metabolic pathways of the analyzed cyanobacterial strains. This analysis combined pathway enrichment (global test) and pathway topology (relative betweenness centrality) methods. The former is used to highlight pathways that include metabolites displaying statistically significant alternations in their relative abundances between compared groups. The latter assesses metabolic pathways by measuring the contribution of the identified compounds to the corresponding metabolic pathway. The assignment of metabolites from the input list to metabolic pathways relied on the cyanobacterium *Synechococcus elongatus* PCC7942 metabolic pathway library available at the Kyoto Encyclopedia of Genes and Genomes (KEGG) database (https://www.genome.jp/kegg/, accessed on 15 December 2023). The results of the pathway analysis were visualized as a scatter plot built in coordinates of −log_10_ (*p*-value) (*Y*-axis) and the impact value reflecting the metabolite’s contribution to a given metabolic pathway (pathway impact, *X*-axis).

## 5. Conclusions and Perspectives

The results of this exploration work suggest that all the extremotolerant and extremophilic cyanobacterial strains studied exhibited distinctions in their primary metabolomes under storage conditions. The metabolic differences between strains may be associated with their origin from extreme habitats and were more pronounced between strains from different extreme environments. Conversely, a group of strains representing heterocystous diazotrophs exhibited certain shared metabolomic characteristics, irrespective of their extreme habitats. This finding suggests that the metabolomic similarities observed among these strains may be indicative of an ancestral heritage, rather than the development of analogous strategies to cope with a stressful environment. Nevertheless, the extremotolerant heterocystous diazotrophic terrestrial and freshwater strains cultured under similar storage conditions showed lower amounts of strain-specifically accumulated primary metabolites than extremophiles from habitats with high salinity and alkalinity. The latter group was highly diverse in the amount of specific primary metabolites. This may indicate the diversity of adaptive mechanisms in which metabolites may be involved in sustaining microbial life in saline and alkaline environments.

The results of this study also highlighted three strains of the genus *Limnospira* which, despite their close evolutionary relationship, displayed considerable metabolome variability and a *Nodularia* strain which among all studied strains was distinguished by the majority of accumulated compounds. Despite the analogous habitats in which the four strains evolved, they may form disparate adaptations to adjust their metabolism to thrive in high-Na^+^ and high-pH conditions. Therefore, these four strains might be a subject worthy of further investigation, specifically with regard to the molecular mechanisms underpinning their high metabolic plasticity. In addition, the following directions are suggested as potential areas for further research in relation to this study: (1) assessing the effect of storage conditions on the metabolism of cyanobacteria and determining the sensitivity of their adaptations to storage conditions, which is of particular interest when considering groups of terrestrial and freshwater strains, especially those that are tolerant to desiccation and exhibit depleted strain-specific metabolite profiles under storage conditions; (2) identification of major strain-specific secondary metabolites in the stored strains, as combining this information with primary metabolomics data will allow the linkages between primary and secondary pathways to be established and their contribution to strain-specific adaptations to be assessed and will also allow the biotechnological potential of each strain to be assessed.

## Figures and Tables

**Figure 1 ijms-26-02201-f001:**
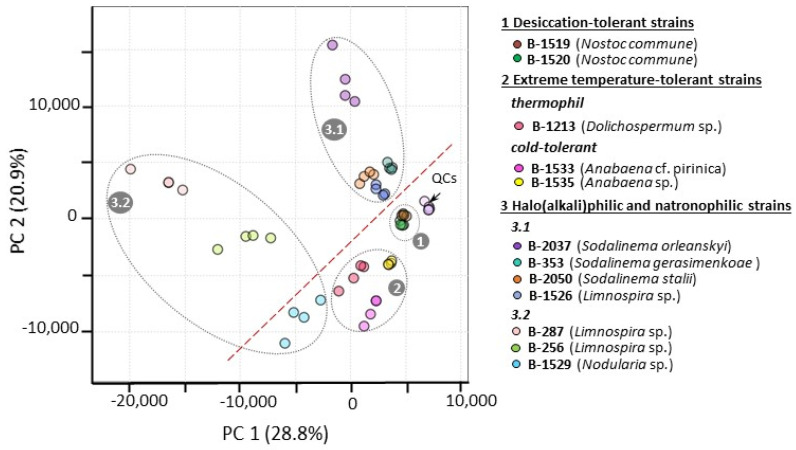
PCA presenting score plot built in coordinates of first two PCs for primary metabolomes of extremophilic and extremotolerant cyanobacteria strains. Input data for the PCA included 530 thermally stable and thermally labile polar metabolites detected by GC-MS and LC-MS in cyanobacterial extracts. Numbers in grey circles and dotted gray line mark clusters formed by metabolomes of cyanobacteria strains adapted to survive in similar extreme environments as follows: desiccation-tolerant strains (1); high-and low-temperature-tolerant strains (2); halo(alkali)philic and natronophilic strains (3) are indicated as two subgroups 3.1 and 3.2 for strains without or with considerable intergroup variability, respectively. The dotted red line separates the groups of strains possessing nitrogen-fixation ability (heterocyst-forming diazotrophs, left) and those lacking this ability (right). QCs—quality control samples. The corresponding loading plot is presented in Appendix A.

**Figure 2 ijms-26-02201-f002:**
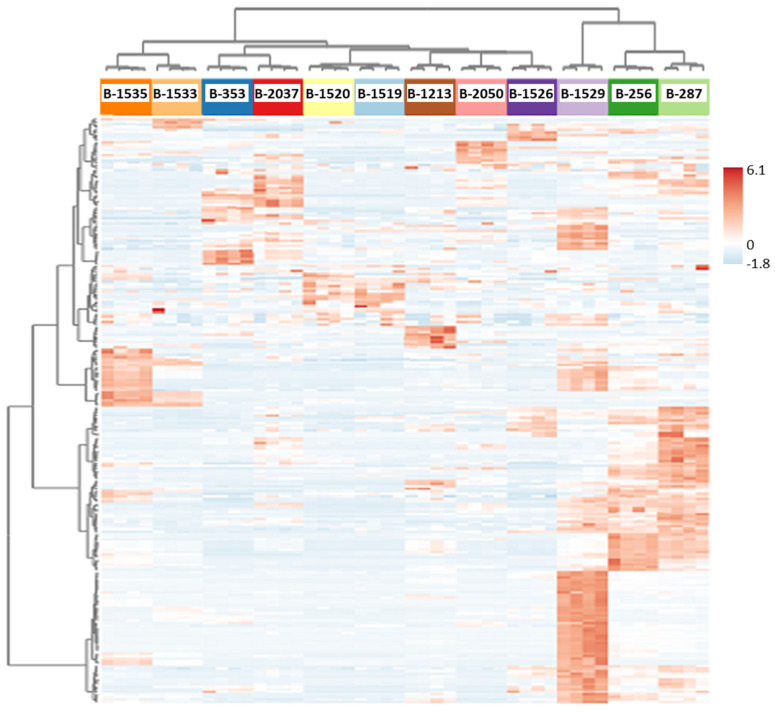
Heatmap representing the overall view of relative abundances of 530 metabolite features (both annotated and nonannotated) detected by GC-MS and LC-MS in polar extracts of twelve studied strains of extremophilic cyanobacteria. Each color marks four samples attributed to the indicated specified cyanobacteria strain. The strain descriptions are given in Table 1 and Figure 1.

**Table 2 ijms-26-02201-t002:** Patterns of major metabolites * associated with desiccation-tolerant cyanobacterial strains *N. commune* B-1520 and B-1519.

Metabolite ^a^	Chemical Class ^b^	TMS Derivative (Feature) ^c^	Strains Tolerant to Desiccation				Method
			B-1520		B-1519		
			FC	*p* ^d^	FC	*p*	
Salicylic acid	Ph	2TMS	14	<0.001			GC-MS
Erythritol	P	4TMS	4.4	0.02	6.5	<0.001	GC-MS
Ethanolamine	A	3TMS	4.2	<0.001			GC-MS

* The metabolites were detected by GC-MS and LC-MS in polar extracts of cyanobacteria cultivated under storage conditions and showed significant differences (FC ≥ 2, *p* ≤ 0.05) in relative content in specified cyanobacterial strains compared to the average content of these metabolites in all extremophilic strains studied. ^a^ Only structurally annotated metabolites are presented. The whole pattern including nonannotated compounds (unknowns) is presented in Appendix A. The metabolites are arranged according to their chemical structure: ^b^ phenolic compound (Ph), polyol (P), amine (A). ^c^ Specific metabolite features representing each metabolite. TMS—trimethylsilyl group. ^d^ The Student’s *t*-test *p*-value was calculated with the false discovery rate (FDR) obtained according to the Benjamini–Hochberg approach and did not exceed the *p*-value threshold 0.05 after applying the FDR correction. In addition, Welch’s *t*-test (*p* ≤ 0.05) and the Mann–Whitney U-test (*p* ≤ 0.01) were used to estimate the significance of the difference between the compared groups for each metabolite reported here and are presented in Appendix A.

**Table 3 ijms-26-02201-t003:** Patterns of major metabolites * associated with each specified cyanobacterial strain B-1213, B-1533 and B-1535 adapted to extreme temperature environment.

Metabolite ^a^	Chem. Class ^b^	Feature ^c^	Strains Tolerant to Extreme High or Low Temperatures						Method
			*Dolichospermum* sp., B-1213 (Thermotolerant)		*Anabaena* cf. *pirinica,* B-1533 (Cold-Tolerant)		*Anabaena* sp., B-1535 (Cold-Tolerant)		
			FC	*t*-Test *p* ^d^	FC	*t*-Test *p*	FC	*t*-Test *p*	
Glycolic acid	CA						2.4	<0.001	LC-MS
Aconitic acid	CA				9	<0.001			LC-MS
3-Dehydroshikimic acid	CA				2.3	0.001	2.3	0.03	LC-MS
Glucose	S	1MEOX, 5TMS (1)			12	<0.001			GC-MS
		1MEOX, 5TMS (2)			8.4	<0.001			GC-MS
Mannitol	P	6TMS					10	<0.001	GC-MS
2C-methyl-*D*-erythritol	P				17	<0.001			LC-MS
Digalacturonic acid	SA				5.1	<0.001	3.8	0.002	LC-MS
Glucose 1-phosphate	SP		4	<0.001					LC-MS
Glucose 6-phosphate	SP				5.9	<0.001			LC-MS
		1MEOX, 6TMS (1)			5.9	<0.001			GC-MS
		1MEOX, 6TMS (2)			6.1	<0.001			GC-MS
Fructose 6-phosphate	SP				6.1	<0.001			LC-MS
		1MEOX, 6TMS			5.1	<0.001			GC-MS
2-Keto-3-deoxy-6-phosphogluconate	SP				6	<0.001			LC-MS
2C-methyl-*D*-erythritol-4-phosphate (MEP)	SP				4.2	0.03	5.4	<0.001	LC-MS
Ribulose 5-phosphate /Xylulose 5-phosphate	SP				3.6	<0.001			LC-MS
Phosphate	P_n_		4	<0.001					LC-MS
Glutamic acid	AA		5.3	<0.001					LC-MS
Ureidosuccinic acid	AAd				9.3	<0.001	9.7	<0.001	LC-MS
NADH	Nuc		4.2	0.008					LC-MS
cGMP	Nuc						5.8	<0.001	LC-MS
ADP-ribose	NucS		27	<0.001					LC-MS
2-Hydroxypyridine	ON	1TMS	5.6	<0.001					GC-MS
Dihydroorotic acid	ON						4.4	<0.001	LC-MS
Nonadecan-1-ol	FAl	1TMS			9.4	<0.001			GC-MS

* The metabolites were detected by GC-MS and LC-MS in polar extracts of cyanobacteria cultivated under storage conditions and showed significant differences (FC ≥ 2, *p* ≤ 0.05) in relative content in specified cyanobacterial strains compared to the average content of these metabolites in all extremophilic strains studied. ^a^ Only structurally annotated metabolites are presented and arranged according to their chemical structure: ^b^ carboxylic acids (CAs), sugars (Ss), polyols (Ps), sugar-derived acids (SAs), sugar phosphates and derivatives (SPs), phosphate (P_n_), amino acids (AAs) and derivatives (AAds), nucleotides (Nucs), sugar nucleotides (NucSs), other N-containing compounds (ONs), fatty alcohol (FAl). ^c^ The features representing each metabolite are specified with characteristic methyloxime (MEOX) and trimethylsilyl (TMS) derivatization groups. The whole pattern including compounds annotated to chemical class and nonannotated compounds (unknowns) is presented in Appendix A. ^d^ The Student’s *t*-test *p*-value was calculated with the FDR obtained according to the Benjamini–Hochberg approach [42]. Welch’s *t*-test (*p* ≤ 0.05) and the Mann–Whitney U-test (*p* ≤ 0.01) additionally applied to estimate the significance of the difference between the compared groups for each metabolite reported here are presented in Appendix A.

**Table 4 ijms-26-02201-t004:** Patterns of major metabolites * associated with halo(alkali)philic and natronophilic cyanobacteria strains from subgroup without appearing considerable intergroup variability (B-2037, B-353, B-2050 and B-1526).

Metabolite ^a^	Chem. Class ^b^	Feature ^c^	*Sodalinema orleanskyi,* B-2037		*S. gerasimenkoae,* B-353		*S. stalii,* B-2050		*Limnospira* sp., B-1526		Method
			FC	*t*-Test ^d^	FC	*t*-Test *p*	FC	*t*-Test *p*	FC	*t*-Test *p*	
3-Hydroxypyruvate	CA		0.34	0.04							LC-MS
Lactic acid	CA	2TMS							3	0.01	GC-MS
3-Hydroxybutyric acid	CA	2TMS							20	<0.001	GC-MS
Isocitric acid	CA		4.9	<0.001			4.9	<0.001			LC-MS
Shikimic acid	CA				4.4	<0.001					LC-MS
2-Phosphoglycolate	CAP		6.1	<0.001							LC-MS
Ribose	S	1MEOX, 4TMS					4.1	<0.001			GC-MS
RI2255 Glucosylglycerol 1	S	6TMS	10	<0.001							GC-MS
RI2310 Glucosylglycerol 2	S	6TMS							8.5	<0.001	GC-MS
Ribonic acid	SA						6.5	<0.001			LC-MS
*D*-Erythronic acid	SA	4TMS					2.8	0.001			GC-MS
Galacturonic acid	SA						13	<0.001			LC-MS
Glycerol	P	3TMS					3.5	<0.001			GC-MS
Sedoheptulose-1,7-biphosphate	SP		5.6	<0.001							LC-MS
Pyroglutamic acid	AA	2TMS							2.7	0.01	GC-MS
NADPH	Nuc						8.6	<0.001			LC-MS
ADP	Nuc		5.7	<0.001							LC-MS
CDP	Nuc		5.8	<0.001							LC-MS
GDP	Nuc		4.9	<0.001							LC-MS
Adenosine	Nuc				41	<0.001					LC-MS
Orotic acid	ON						5.0	<0.001			LC-MS
*trans*-9-Hexadecenoic acid	FA	1TMS	9.1	<0.001							GC-MS
Phosphoric acid	IO	3TMS	2.2	0.03							GC-MS
Chloride	IO		2.6	<0.001	2.5	<0.001					LC-MS

* The metabolites were detected by GC-MS and LC-MS in polar extracts of cyanobacteria cultivated under storage conditions and showed significant differences (FC ≥ 2, *p* ≤ 0.05) in relative content in specified cyanobacterial strains compared to the average content of these metabolites in all extremophilic strains studied. ^a^ Only structurally annotated metabolites presented and arranged according to their chemical structure: ^b^ carboxylic acids (CAs) and their phosphate derivatives (CAPs), sugars (Ss), sugar-derived acids (SAs), polyols (Ps), sugar phosphates (SPs), amino acids (AAs), nucleotides and nucleosides (Nucs), other N-containing compounds (ONs), fatty acids (FAs), inorganic ions (IOs). ^c^ The features representing each metabolite are specified with characteristic methyloxime (MEOX) and trimethylsilyl (TMS) derivatization groups. The whole pattern including compounds annotated to chemical class and unknowns is presented in Appendix A. ^d^ The Student’s *t*-test *p*-value was calculated with the FDR obtained according to the Benjamini–Hochberg approach [42]. Welch’s *t*-test (*p* ≤ 0.05) and the Mann–Whitney U-test (*p* ≤ 0.01) additionally applied to estimate the significance of the difference between the compared groups are presented in Appendix A.

**Table 5 ijms-26-02201-t005:** Patterns of major metabolites * associated with specified haloalkaliphilic and natronophilic cyanobacteria strains B-1529, B-287 and B-256 demonstrating considerable intergroup metabolome variability.

Metabolite ^a^	Chem. Class ^b^	Feature ^c^	*Nodularia* sp., B-1529		*Limnospira* sp., B-287		*Limnospira* sp., B-256		Method
			FC	*t*-Test *p* ^d^	FC	*t*-Test *p*	FC	*t*-Test *p*	
3-Phosphoglyceric acid	CAP				14	<0.001			LC-MS
2-Phosphoglyceric acid	CAP				18	<0.001			LC-MS
Phosphoenolpyruvic acid	CAP				27	<0.001			LC-MS
Fructose	S	1MEOX, 5TMS (1)	32	<0.001					GC-MS
		1MEOX, 5TMS (2)	35	<0.001					GC-MS
Mannose	S	1MEOX, 5TMS	17	<0.001					GC-MS
Galactose	S	1MEOX, 5TMS	17	<0.001					GC-MS
Glucose	S	1MEOX, 5TMS	14	<0.001					GC-MS
α,α-Trehalose	S	8TMS			26	<0.001			GC-MS
*D*-galactonic acid	SA		13	<0.001					LC-MS
1-Deoxyxylulose 5-phosphate	SP						13	<0.001	LC-MS
2-Deoxyribose 5-phosphate	SP						12	<0.001	LC-MS
Sedoheptulose 7-phosphate	SP		11	<0.001					LC-MS
Glucosamine 6-phosphate	SP						14	<0.001	LC-MS
Alanine	AA	2TMS	10	<0.001					GC-MS
Arginine	AA		17	<0.001					LC-MS
Argininosuccinate	AA		11	<0.001					LC-MS
Aspartic acid	AA		11	<0.001					LC-MS
		3TMS	25	<0.001					GC-MS
Serine	AA		25	<0.001					LC-MS
Threonine	AA		46	<0.001					LC-MS
Valine	AA	2TMS	16	<0.001					GC-MS
Isoleucine	AA	2TMS	25	<0.001					GC-MS
Glycine	AA		15	<0.001					LC-MS
		2TMS	24	<0.001					GC-MS
Phenylalanine	AA		43	<0.001					LC-MS
Tryptophan	AA		33	<0.001					LC-MS
Tyrosine	AA	2TMS	30	<0.001					GC-MS
		3TMS	12	<0.001					GC-MS
Methionine	AA		80	<0.001					LC-MS
Glutamine	AA		34	<0.001					LC-MS
Cytidine	Nuc		14	<0.001					LC-MS
Uridine	Nuc		21	<0.001					LC-MS
Guanosine	Nuc		22	<0.001					LC-MS
2′-Deoxyguanosine	Nuc		19	<0.001					LC-MS
ATP	Nuc				10	<0.001			LC-MS
GTP	Nuc				10	<0.001			LC-MS
3-Ureidopropionic acid	ON		74	<0.001					LC-MS
Uric acid	ON		42	<0.001					LC-MS
Isovaleryl-CoA	CoAt		13	<0.001					LC-MS
Acetoacetyl-CoA	CoAt		13	<0.001					LC-MS
*S*-acetyl-CoA	CoAt				11	<0.001	7.0	<0.001	LC-MS
*trans*-9-Octadecenoic acid	FA	1TMS	18	<0.001					GC-MS
Linoleic acid	FA	1TMS	18	<0.001					GC-MS
Oleic acid	FA	1TMS	30	<0.001					GC-MS
*cis*-9-Hexadecenoic acid	FA	1TMS	14	<0.001					GC-MS
*delta3*-isopentenyl pyrophosphate	PP				10	<0.001			LC-MS
Dimethylallylpyrophosphate	PP				10	<0.001			LC-MS

* Metabolites were detected by GC-MS and LC-MS in polar extracts of cyanobacteria cultivated under storage conditions and showed significant differences (FC ≥ 10, *p* ≤ 0.001) in relative content in specified cyanobacterial strains compared to the average content of these metabolites in all extremophilic strains studied. ^a^ Only structurally annotated metabolites exhibiting a considerable increase (≥10-fold) in abundance are presented and arranged according to their chemical structure: ^b^ carboxylic acid phosphate derivatives (CAPs), sugars (Ss), sugar-derived acids (SAs), sugar phosphates (SPs), amino acids (AAs), nucleotides and nucleosides (Nucs), other N-containing compounds (ONs), CoA thioesters (CoAts), fatty acids (FAs), prenyl phosphates (PPs). ^c^ Metabolite features representing each metabolite are specified with characteristic MEOX and TMS derivatization groups. The whole pattern including all compounds (annotated and nonannotated) demonstrating ≥ 2-fold difference (Student’s *t*-test *p* (FDR-adjusted) ≤ 0.05, Welch’s *t*-test *p* ≤ 0.05, Mann–Whitney U-test *p* ≤ 0.01) in abundance compared to their average levels is presented in Appendix A. ^d^ The Student’s *t*-test *p*-value was calculated with the FDR obtained according to the Benjamini–Hochberg approach [42].

**Table 6 ijms-26-02201-t006:** Patterns of major differentially expressed metabolites * in heterocyst-forming and non-heterocystous strains.

Metabolite ^a^	Chem. Class ^b^	Feature ^c^	FC	*t*-Test *p* ^d^	Method
Metabolites with higher relative content in heterocyst-forming compared to non-heterocystous strains					
Salicylic acid	CA	2TMS	18	0.009	GC-MS
Glucose	S		5.5	0.030	LC-MS
Mannose	S	1MEOX, 5TMS	6.8	0.031	GC-MS
Mannitol	P	6TMS	5.0	0.031	GC-MS
2C-methyl-*D*-erythritol	P		43	0.015	LC-MS
Fructose 6-phosphate	SP		5.3	0.010	LC-MS
Glucose 6-phosphate	SP		5.0	0.011	LC-MS
Sedoheptulose 7-phosphate	SP		28	0.002	LC-MS
2C-methylerythritol 4-phosphate	SP		31	<0.001	LC-MS
2-keto-3-deoxy-6-phosphogluconate	SAP		5.1	0.015	LC-MS
Tyrosine	AA	2TMS	7.2	0.027	GC-MS
Argininosuccinate	AA		28	0.002	LC-MS
Ureidosuccinic acid	AA		37	0.008	LC-MS
Nonadecan-1-ol	FAl	1TMS	5.7	0.007	GC-MS
Metabolites with lower relative content in heterocyst-forming compared to non-heterocystous strains					
2-phosphoglycolic acid	CAP		14	<0.001	LC-MS
2-phosphoglyceric acid	CAP		39	0.009	LC-MS
3-phosphoglyceric acid	CAP		24	0.005	LC-MS
Phosphoenolpyruvic acid	CAP		25	0.024	LC-MS
α,α-Trehalose	S	8TMS	65	0.017	GC-MS
RI2255 Glucosylglycerol	S		>100	<0.001	GC-MS
Fructose-1,6-diphosphate	SP		27	0.001	LC-MS
Sedoheptulose-1,7-biphosphate	SP		9.4	0.001	LC-MS
Adenylosuccinic acid	Nuc		6.5	0.004	LC-MS
GTP	Nuc		17	0.002	LC-MS
ATP	Nuc		22	0.001	LC-MS
CTP	Nuc		5.6	<0.001	LC-MS
UTP	Nuc		7.2	0.004	LC-MS
GDP	Nuc		5.0	0.001	LC-MS
ADP	Nuc		6.6	0.002	LC-MS
dTTP	Nuc		5.5	0.013	LC-MS
ADP-glucose	NucS		7.4	0.019	LC-MS
S-acetyl CoA	CoAt		10	0.016	LC-MS
*trans*-9-Hexadecenoic acid	FA	1TMS	5.7	0.008	GC-MS
(*2E*)-4-hydroxy-3-methylbut-2-en-1-yl diphosphate (HMBDP)	PP		8.9	0.003	LC-MS
*∆*^3^-isopentenyl pyrophosphate	PP		10	0.001	LC-MS

* GC-MS- and LC-MS-detected polar metabolites from extracts of cyanobacteria cultivated under storage conditions showing significant differences (FC ≥ 5, *p* ≤ 0.05) in relative content in heterocyst-forming (B-1520, B-1519, B-1213, B-1533, B-1535, B-1529; *n* = 24) compared to non-heterocystous (B-353, B-2050, B-2037, B-1526, B-287, B-256; *n* = 24) extremophilic strains studied. ^a^ Only structurally annotated metabolites exhibiting a considerable difference (≥5-fold) in abundance between compared groups are presented and arranged according to their chemical structure: ^b^ carboxylic acids (CAs) and their phosphate derivatives (CAPs), sugars (Ss), polyols (Ps), sugar phosphates (SPs), sugar-derived acids (SAs) and their phosphate derivatives (SAPs), amino acids (AAs), nucleotides and nucleosides (Nucs), sugar nucleotides (NucSs), CoA thioesters (CoAts), fatty acids (FAs) and their alcohols (FAls), prenyl phosphates (PPs). ^c^ Metabolite features representing each metabolite are specified with characteristic TMS and MEOX derivatization groups. The whole pattern including all compounds (annotated and nonannotated) demonstrating ≥ 2-fold difference (Student’s *t*-test *p* (FDR-adjusted) ≤ 0.05) in abundance is presented in Appendix A. ^d^ The Student’s *t*-test *p*-value was calculated with the FDR obtained according to the Benjamini–Hochberg approach [42].

**Table 7 ijms-26-02201-t007:** Primary metabolites accumulated in specific cyanobacterial strains and the main metabolic processes in which they may be involved to maintain the life of extremotolerant and extremophilic strains under long-term storage conditions *.

Tolerance Group	Species Name, IPPAS ID, Extreme Environment	Storage Medium, t °C, Storage Period	Strain-Specifically Accumulated Metabolites	Metabolic Processes
Desiccation-tolerant	*Nostoc commune* B-1520, heterocystous diazotroph, terrestrial macrocolony	BG-11 without nitrogen, 22 °C, 3 months	Salicylic acid; erythritol	Production of EPS, osmoprotection, component of EPS
	*Nostoc commune* B-1519, heterocystous diazotroph, terrestrial macrocolony	BG-11 without nitrogen, 22 °C, 3 months	Erythritol	Osmoprotection, component of EPS
High-temperature-tolerant	*Dolichospermum* sp. B-1213, heterocystous diazotroph, hot springs	BG-11 without nitrogen, 22 °C, 8 months	ADP-ribose, NADH; glucose-1-P, glutamate	Protein ADP-ribosylation; glycogen degradation and redirected carbon flux via glycolysis towards amino acid synthesis
Low-temperature-tolerant up to 10–11 °C	*Anabaena* cf. pirinica B-1533, heterocystous diazotroph, high-rate cold water flow	No. 6 without nitrogen, 22°, 3 months	Hexoses, hexose phosphates, fatty alcohol; glycolysis and pentose phosphate intermediates; ureidosuccinate; 2C-methyl-*D*-erythritol, MEP	Production of EPS and glycolipids; enhanced carbon metabolism via glycolysis and pentose phosphate pathways; de novo pyrimidine biosynthesis; isoprenoid biosynthesis
	*Anabaena* sp. B-1535, heterocystous diazotroph, high-rate cold water flow	No. 6 without nitrogen, 22 °C, 3 months	Glycolate; MEP	Photorespiration; isoprenoid biosynthesis
Haloalkaliphilic and natronophilic	*Sodalinema orleanskyi* B-2037, saline–alkaline lake	S (pH 9.0–9.5) 32 °C, 3 weeks, 22 °C, 2 months	Glucosylglycerols; glycolate-2P	Osmoregulation; photorespiration
	*Sodalinema gerasimenkoae* B-353, saline–alkaline lake	S (pH 9.0–9.5) 27 °C, 3 weeks, 22 °C, 2 months	Shikimate; adenosine	Aromatic amino acids, phenolic compounds; not well understood
	*Sodalinema stalii* B-2050, halophilic, coastal shoals	ASNIII, (pH 7.5) 27 °C, 3 weeks, 22 °C, 2 months	Glycerol, galacturonic acid; orotic acid; NADPH	EPS production; de novo pyrimidine biosynthesis; low Calvin cycle activity
	*Limnospira* sp. B-1526, soda lake	Zarrouk, (pH 9) 32 °C, 6 weeks	Glucosylglycerol; 3-hydroxybutyrate	Osmoprotection; PHB synthesis
	*Nodularia* sp. B-1529, heterocystous diazotroph, soda lake	Zarrouk, 27 °C, 3 weeks, 22 °C, 3 weeks	Amino acids; compatible solutes (hexoses, sucrose, glucosylglycerol)	Nodularin biosynthesis (protein protection from oxidative damage or N-storage polymer?); osmoprotection
	*Limnospira* sp. B-287, origin is not known, haloalkaliphile, natronophile	Zarrouk, (pH 9) 32 °C, 6 weeks	ATP, CoA-thioesters, isopentenyl-PP; trehalose	Biosynthetic processes (polyketide, fatty acid, PHB, isopropanoids); osmoprotection
	*Limnospira* sp. B-256, soda lake	Zarrouk, (pH 9) 32 °C, 6 weeks	Deoxypentoses; deoxy-ribonucleotides; UDP-*N*-acetylglucosamine	De novo nucleotide synthesis; cell wall biogenesis

* Reduced light supply up to light intensity 50 µmol photons m^−2^ s^−1^ and no additional CO_2_ supply.

## Data Availability

All relevant data are available within the article and Appendix A.

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
