# Peer review of "Strain-Specific Features of Primary Metabolome Characteristic for Extremotolerant/Extremophilic Cyanobacteria Under Long-Term Storage"

_ijms, 2025, doi:10.3390/ijms26052201_

Round 1

Reviewer 1 Report

Comments and Suggestions for Authors

Review of IJMS-3389849

This large manuscript describes the metabolic profiling of 12 extremophile cyanobacterial strains which had been taken from culture collections where they had been maintained on strain-specific media, under conditions of low light and CO2. The study is described as an “exploratory” study. In particular it is noted that the strains are unialgal but not axenic, so bacteria are also present. The standard of English needs some editing but on the whole is acceptable. The manuscript is lengthy and complex but in general the organisation and logical flow is OK. The methods appear to be reasonable and well executed. On a minor note, statistically, where there are many tests of significance, one would query whether a significance cutoff of p<0.05 is stringent enough.

The primary question posed in the Abstract is whether these strains “retain the primary metabolome features associated with their unique adaptions”. Presumably, this question implies that these strains may have lost their extremophile adaptions due to long term culture in collections, or because the pathways needed for extremophilic culture were not induced under the conditions of culture.

The authors found that indeed there were strain-specific metabolites, especially in organisms derived from highly saline and alkaline environments. An attempt was made to tie the strain-specific metabolites to particular pathways that might be of relevance for extremophile survival mechanisms.

I have mixed feelings about this work. On the one hand, the amount of data generated is large. If such data is robust (for example, truly representative of the biology of these strains and not influenced by the specific culture conditions) then such a data set is likely to be of broad interest, even if there is no underlying hypothesis that ties the data together.

However, the weaknesses of the experimental design and the analysis are significant.

First, not surprisingly, extremophiles need to be cultured in different media (see Table 1) and these different media were used for growth prior to harvesting and analysis. One would expect that the metabolite profile will necessarily reflect the growth media to at least some extent. How different would the metabolite profiles be if the cells had been grown in different media? This point does not appear to be addressed.

Second, how can one tell, from the data presented, how different the metabolite profiles of the culture collection strains might be from the original wildtype strains (of which we now have no record)? Given the lack of this data, how could we tell whether there had been any divergence over the years of culture? One way to try to deal with this is to grow the strains under extremophilic and stressed conditions to see whether they survive as well as when they were first isolated (assuming that data is available). Even without metabolic data, this would suggest whether these strains had lost capabilities over the time of culture.

Third, many of these strains that occur in similar environments are (not surprisingly) closely related (e.g. B-1526, B-287, B-256). This close evolutionary relationship is as likely to be the cause of similar metabolite profiles as the conditions under which they are grown. How

then, can we say that “halophiles and natrophiles” share a similar metabolic profile when at least 3 of the “halophiles and natrophiles” are close relatives? As the authors note, diazotrophs cluster separately in Fig 1 and this is not simply because they have developed similar strategies to cope with an extreme environment – rather, it reflects their ancient genetic legacy. This is acknowledged by the authors in the Discussion (lines 568-577) but the problem this causes for the attribution of metabolites is not discussed specifically. Therefore the clustering analysis cannot tease apart the various potential influences, some of which are of interest for understanding extremophile adaption, and others which are not.

As many strategies for extremophile adaption have been described in the literature, pathway analysis can be useful in linking the strain-specific metabolites to known extremophile adaptions, such as compatible solutes. Where this has not been possible, it suggests that new, previously undiscovered adaptions may be present. The trouble is, that such tantalising possibilities are not followed up or validated. Therefore this study throws up several interesting directions for further research, but no hard data or follow up experiments to really support them. The result is indeed an “exploratory” study, a large and unwieldy data repository with many hints but few conclusions that can be drawn.

The question of whether it is desirable to publish this work rests on the trade-offs between providing, on the one hand, a data set that would be of interest to extremophile research, and on the other, publishing a work which lacks a cogent hypothesis, nor any kind of follow- up or validating experiments to enable the reader to draw conclusions. In my view, there is a place for such exploratory studies, but whether such studies suit the aims of the journal is and editorial decision that is outside the purview of the review process.

If publication is undertaken, the authors need to, at the very least, draw attention to and address the points raised above. Similarly, where there is the possibility that key strain- specific metabolites might be contributed by bacterial contaminants, this possibility should be addressed, for example by looking in the literature at known bacterial sources of these metabolites.

Comments on the Quality of English Language

The grammar needs correction in many places not least to remove ambiguities and difficulty in comprehension.  There are some typos which I assume will be dealt with if the manuscript is typset.

Author Response

We thank the Reviewer for thoughtful review and highly appreciate the valuable comments and suggestions to improve the manuscript. Following these advices, we have corrected statements and supplemented the manuscript with relevant information in corresponding sections.

Comment 1: This large manuscript describes the metabolic profiling of 12 extremophile cyanobacterial strains which had been taken from culture collections where they had been maintained on strain-specific media, under conditions of low light and CO2. The study is described as an “exploratory” study. 

In particular it is noted that the strains are unialgal but not axenic, so bacteria are also present. The standard of English needs some editing but on the whole is acceptable. The manuscript is lengthy and complex but in general the organization and logical flow is OK. The methods appear to be reasonable and well executed. 

Response: Thank you very much for your appreciation of the organization and methodological approach employed in our work.

Remark 1: On a minor note, statistically, where there are many tests of significance, one would query whether a significance cutoff of p<0.05 is stringent enough.

Response: We thank the Reviewer for highlighting this question. For metabolomic studies, a p-value≤0.05 (a predetermined probability threshold) is typically advised when employing a t-test to reject the null hypothesis (H0) that the mean values of two groups are equal [Vinaixa et al., 2012; Chen et al., 2022; Xi et al., 2014]. However, given the substantial number of metabolites present in any given biological sample, and the separate t-test conducted for each individual metabolite without considering the tests for other metabolites, this approach can result in a large number of false positive results. To address this, a multiple comparison procedure with a reduced cutoff p-value is employed to control the overall error caused by employing all the t-tests together. Among the multiple testing methods, Benjamini-Hochberg approach also known as False Discovery Rate (FDR) is considered to be one of the most effective [Vinaixa et al., 2012; Chen et al., 2022], and it was employed in our study to mitigate the false positives in significant results obtained.

Accordingly, the following text and reference (highlighted with bold font) were added to Method section (Statistical analysis) (lines 1081-1085):

“For FC and t-test p presented at tables and figures the thresholds values were as follows: minimal value of FC was 2 and maximal of p-value was 0.05 as typically recommended for metabolomics studies [Vinaixa et al., 2012; Xi et al., 2014]. The reliability of the observed FCs was assessed by adjusting p-value with false discovery rate (FDR) estimated with Benjamini-Hochberg method.

However even if the two sample group mean values are found to be different (t-test p-value (FDR adjusted)≤0.05), there may be substantial overlap between the samples of the two groups. As we compare two groups of differing sample sizes (each strain (n=4) versus a group obtained by pooling samples (n=44) of other 11 strains) it is important to take into account the group different variances and the potential for the non-normal data distribution. To this end, the revised manuscript has been supplemented with Welch's t-test (test for unequal variances) and Mann-Whitney U-test (test for non-normally distributed data sets) [Vinaixa et al., 2012; Xi et al., 2014], in order to validate the significance of the observed differences in the abundances of metabolites detected by the Volcano analysis (Student's t-test). The threshold p-values for the tests were as follows: the maximum p-value for the Welch's t-test was 0.05 and for the Mann-Whitney U-test was 0.01. The rationale behind selecting a p-value cutoff of 0.01 but not 0.05 for the Mann-Whitney U-test is explained that this non-parametric test is subject to less stringent assumptions in comparison with the t-tests. Thus, difference in metabolite abundances with p-values less than 0.05 (Student’ and Welch’s tests) and 0.01 (U-test) were considered statistically significant. By implementing the aforementioned p-value cutoffs, the numbers of significant metabolites identified in our study are now accurately presented in the revised manuscript and Supplementary 2 (these amendments are highlighted with yellow). Additionally, we have added combined violin and box plots for significant strain-specific metabolites to the Supplementary 2 Figures S2-2 – 13 and have added the following information to the revised manuscript:

Method section (Statistical analysis) (lines 1085-1091):

“As the variance of the groups compared was different and the data may not be normally distributed, the significance of the observed differences in the abundances of metabolites detected by the Volcano analysis was additionally assessed using Welch's t-test and Mann-Whitney U-test [Vinaixa et al., 2012; Xi et al., 2014]. The threshold p-values for the tests reported in the tables were as follows: the maximum p-value for the Welch's t-test was 0.05 and for the Mann-Whitney U-test was 0.01.”

Results section (lines 277-280):

“Further, taken in consideration different variances of compared groups and expected non-normally distributed data, the significance of difference between groups in abundance of metabolites indicated by Volcano analysis was additionally estimated by Welch’s t-test and Mann-Whitney U-test. This approach resulted in four metabolite features…”

Tables 2–5 footnotes (lines 297-299, 335-337, 401-404, 466)

“In addition, Welch's t-test (p≤0.05) and Mann-Whitney U-test (p≤0.01) were used to estimate the significance of the difference between the compared groups for each metabolite reported here and are presented in Supplementary 2, Table…”

Referenced literature

Vinaixa, M., Samino, S., Saez, I., Duran, J., Guinovart, J. J., & Yanes, O. (2012). A Guideline to Univariate Statistical Analysis for LC/MS-Based Untargeted Metabolomics-Derived Data. Metabolites2(4), 775–795. https://doi.org/10.3390/metabo2040775

Chen, Y., Li, E. M., & Xu, L. Y. (2022). Guide to Metabolomics Analysis: A Bioinformatics Workflow. Metabolites12(4), 357. https://doi.org/10.3390/metabo12040357

Xi, B., Gu, H., Baniasadi, H., & Raftery, D. (2014). Statistical analysis and modeling of mass spectrometry-based metabolomics data. Methods in molecular biology (Clifton, N.J.)1198, 333–353. https://doi.org/10.1007/978-1-4939-1258-2_22

Comment 2: The primary question posed in the Abstract is whether these strains “retain the primary metabolome features associated with their unique adaptions”. Presumably, this question implies that these strains may have lost their extremophile adaptions due to long term culture in collections, or because the pathways needed for extremophilic culture were not induced under the conditions of culture.

Response: We agree with the Reviewer’s comment, indeed this primary question which we posed in Abstract “whether these strains retain the primary metabolome features associated with their unique adaptations” includes the both thoughts expressed by the Reviewer. On the one hand, due to long term storage conditions, which are not considered extreme conditions for temperature and desiccation tolerant strains, their unique adaptations are not induced and therefore it is conceivable that the unique adaptations might be lost as unclaimed over time. But this is not a case for those adaptations that once arose as environmentally induced changes in the phenotype of the organisms and were consistently favourable to the organism in their natural environment. Those adaptations may lose sensitivity to the initial environmental factor and lead to the constitutive expression of the traits (this thought was described in Discussion, lines 618-627 in revised manuscript). These traits are not considered to be lost under long storage conditions.

In the case of extremophilic cyanobacteria (halo-(alkali -and natrono)philic strains), their adaptations are based on constitutive expressed traits associated with multiple rearrangements in their metabolomes which made the organisms to be obligately dependent on high Na concentration and pH, for their growth and development.

To identify those strain-specific constitutive expressed primary metabolome features in the stored strains associated with formation of the strain unique adaptions was the objective of our study. In order to clarify the focus of our study, title of revised manuscript was changed as follows:

“Strain-specific features of primary metabolome characteristic for extremotolerant/extremophilic cyanobacteria under long-term storage”

and the text in Abstract (line 21) was changed accordingly (the change is highlighted with bold):

“However, it remains unclear whether these strains maintain constitutively expressed primary metabolome features associated with their unique adaptations.”

Comment 3: The authors found that indeed there were strain-specific metabolites, especially in organisms derived from highly saline and alkaline environments. An attempt was made to tie the strain-specific metabolites to particular pathways that might be of relevance for extremophile survival mechanisms.

I have mixed feelings about this work. On the one hand, the amount of data generated is large. If such data is robust (for example, truly representative of the biology of these strains and not influenced by the specific culture conditions) then such a data set is likely to be of broad interest, even if there is no underlying hypothesis that ties the data together.

Response: We thank the Reviewer for his/her assessment of the significance of our exploratory study. We have no doubts about the validity of the data presented. The results of the study have been verified using three statistical tests, and violin and box plots for the most different and statistically significant strain-specific metabolites have been added to the Supplementary Information 2 (Figures S2-2 – 13) to indicate comparable data distributions. The strain culturing conditions described in the Table 1 are reliable and are used on a daily basis to maintain corresponding strains of the IPPAS microalgae and cyanobacteria collection for long-term storage. Thus, our data presented herein is truly representative of the biology of these strains under conditions of long-term storage cultivation. The following clarifying text was added in the sentences to the Method section (the changes are highlighted with bold font, lines 954-955):

“For the experiments, the strains were grown in 350-ml Erlenmeyer flasks containing 150–200 ml of appropriate liquid medium under conditions (reduced light and an absence of additional CO2 supply) which are used IPPAS collection for long-term maintenance of the studied strains (Table 1).

Remark 2: However, the weaknesses of the experimental design and the analysis are significant.

 First, not surprisingly, extremophiles need to be cultured in different media (see Table 1) and these different media were used for growth prior to harvesting and analysis. One would expect that the metabolite profile will necessarily reflect the growth media to at least some extent. How different would the metabolite profiles be if the cells had been grown in different media? This point does not appear to be addressed.

Response: We agree with the Reviewer that a strain’ metabolite profile will necessarily reflect the growth media. However, it was out of score of our study to investigate how culture media will impact strains metabolite profiles. This subject might be of a focus for the next investigation. Nevertheless, in our current study we focused on metabolite profile of strains which were grown on the well-established media which provided optimal growth and development of the strains. The references to the strain media description are indicated in Table 1. The media are used on routine basis for the studied herein strains to maintain the corresponding strains of the IPPAS collection. The long-term storage of the strains grown on media optimal for their growth was maintained by reduced light supply and without additional CO2 supply. The limited light and CO2 supply provided the suboptimal conditions to ensure long-term storage of the strains in a metabolically active state. Accordingly, the sentence in the Introduction section was changed as follows (the changes are highlighted with bold font, lines 103-106):

“The strains, isolated from different extreme habitats, were cultivated in strain specific media under conditions suboptimal for growth, which were provided by reduction of light and the absence of additional CO2 supply. These cultivation conditions guaranteed the long-term maintenance (storage) of the strains in a metabolically active state [29].”

Remark 3: Second, how can one tell, from the data presented, how different the metabolite profiles of the culture collection strains might be from the original wildtype strains (of which we now have no record)? Given the lack of this data, how could we tell whether there had been any divergence over the years of culture? One way to try to deal with this is to grow the strains under extremophilic and stressed conditions to see whether they survive as well as when they were first isolated (assuming that data is available). Even without metabolic data, this would suggest whether these strains had lost capabilities over the time of culture.

Response: We thank the Reviewer for arising this question. Our pilot (exploratory) study on the primary metabolomics of extremophilic cyanobacterial strains was designed based on horizontal comparisons (i.e., between-strain comparisons) of metabolite profiles of twelve strains from diverse extreme locations after long-term storage. The storage conditions of tolerant strains were devoid of the environmental pressure (stress) to which the strains have been adapted. Conversely, the storage conditions for extremophilic strains, which are halo-(alkali -and natrono)philic strains, were almost identical to their natural environment with respect to adaptation to high salinity and alkalinity. In the case of the tolerant strains, we used a comparison of each strain with other strains, expecting to find metabolites that are mainly characteristic to the strain even in the absence of environmental pressure. This may indicate metabolites that are constitutively expressed in the strain and thus may be related to the initial unique adaptation of the strains. Indeed, this approach allowed us to find highly specific accumulation of salicylic acid and erythritol in desiccation-tolerant strains B-1520 and B-1519, respectively (Table 2). On the other hand, extremophilic halo-(alkali -and natrono)philic strains displayed more richer strain-specific metabolite patterns. The extremophiles have adapted to thrive in high salt or/and soda environments and have a greater number of constitutively expressed genes enabling mechanisms of evolved acclimatization [Pade et al., 2014]. Their final metabolite products are expected to be synthetized in strains during their cultivation under conditions of long-term storage. Therefore, the differences obtained in their metabolite profiles might be associated with the strain-specific acclimation mechanisms. The rationales behind the study design were explained in the beginning of Discussion section (lines 610-627 in the revised manuscript).  

Answering to the second part of the Reviewer’s question we would note that we plan to conduct experiments to investigate the metabolome of these strains under stress conditions and we have data on the survival of our strains under corresponding extreme conditions after several years of laboratory cultivation:

  1. Haloalkaliphilic strains of genus Sodalinema, orleanskyi IPPAS B-2037 and S. gerasimenkoae B-353 are able to grow in hypersaline soda medium M (composition g/L: Na2CO3 – 79.5, NaHCO3 – 21.0, KCl –2.0, K2HPO4·3H2O – 0.5, KNO3 – 2.0, Na2SO4 – 1.4, FeCl3 – 0.0003, EDTA – 0.0005, modified trace element solution (TES) A5+Co – 1 mL/L and pH 10.0) even after cultivation in less extreme S medium (composition g/L: NaHCO3 – 16.8, NaCl – 30.0, KCl – 1.0, K2HPO4·3H2O – 0.5, NaNO3 – 2.5, K2SO4 – 1.0, MgSO4·7H2O – 0.1, CaCl2 – 0.04, FeSO4 – 0.01, EDTA – 0.08, modified TES A5+Co – 1 mL/L and pH 9.0–9.5) for 9 years (strain B-2037) and 25 years (strain B-353). This experiment was described in Samylina et al., 2021 https://doi.org/10.1093/femsec/fiab104.
  2. Cold-tolerant strains of Anabaena IPPAS B-1533 and IPPAS B-1535 are stored between passages on agar slants at +12°C for 5-7 months and restore their growth well at room temperature. Many other cyanobacteria do not survive after long storage at +12°C.
  3. Thermotolerant strain IPPAS B-1213 (isolated in 2012) revealed that its optimum growth temperature is at 35-38 °C, at 32 °C the growth is much weaker and the culture survive at 41 °C at least for 6 days (Bozieva et al., 2023). The growth optimum of this strain is mentioned in the Results section (line 128).

Referenced literature

Pade, N., & Hagemann, M. (2014). Salt acclimation of cyanobacteria and their application in biotechnology. Life (Basel, Switzerland)5(1), 25–49. https://doi.org/10.3390/life5010025

Samylina, O. S., Sinetova, M. A., Kupriyanova, E. V., Starikov, A. Y., Sukhacheva, M. V., Dziuba, M. V., & Tourova, T. P. (2021). Ecology and biogeography of the 'marine Geitlerinema' cluster and a description of Sodalinema orleanskyi sp. nov., Sodalinema gerasimenkoae sp. nov., Sodalinema stali sp. nov. and Baaleninema simplex gen. et sp. nov. (Oscillatoriales, Cyanobacteria). FEMS microbiology ecology97(8), fiab104. https://doi.org/10.1093/femsec/fiab104

Bozieva, Ayshat & Khasimov, Makhmadyusuf & Voloshin, Roman & Sinetova, Maria & Kupriyanova, Elena & Zharmukhamedov, Sergey & Dunikov, Dmitry & Tsygankov, Anatoly & Tomo, Tatsuya & Allakhverdiev, Suleyman. (2022). New cyanobacterial strains for biohydrogen production. International Journal of Hydrogen Energy. 10.1016/j.ijhydene.2022.11.198.

Remark 4: Third, many of these strains that occur in similar environments are (not surprisingly) closely related (e.g. B-1526, B-287, B-256). This close evolutionary relationship is as likely to be the cause of similar metabolite profiles as the conditions under which they are grown. How then, can we say that “halophiles and natrophiles” share a similar metabolic profile when at least 3 of the “halophiles and natrophiles” are close relatives? 

Response: We acknowledge the Reviewer's position that it is reasonable to hypothesize that close relatives would always share many metabolome characteristics. However, the results of our study do not indicate that this assumption is always true.

In this study, halophiles and natronophiles are represented by three strains of the genus Limnospira, three strains of the genus Sodalinema and one strain of the genus Nodularia. Sodalinema and Limnospira are phylogenetically quite distant genera of order Oscillatoriales (see Supplementary 1, Figure S1-2) and the heterocystous Nodularia (order Nostocales) is even more distant. To provide an idea how phylogenetical distant or close the strains studied here are we have added the corresponding information (maximum likelihood phylogenetic trees based on partrial 16S rRNA gene sequences) as a Supplementary information 1. 

In response to the query posed by the Reviewer the following text was added to the Discussion section (lines 638-663):

“In addition, the present work has yielded several noteworthy outcomes concerning halo-(alkali- and natrono)philic strains. Firstly, as would be expected, some phylogenetically close strains, e.g. Sodalinema strains, have been shown to share certain metabolome characteristics under long-term storage, despite their initially different ecological environments (Table 1). This is evidenced by the separation of the strains in a cluster in the PCA model (Figure 1). Thus, Sodalinema stalii B-2050, a strain that is only halophilic, but not alkaliphilic and natronophilic, appeared to be in the same cluster as other haloalkaliphilic and natronophilic Sodalinema strains (Figure 1, Cluster 3.1, Supplementary 1, Figure S1-2). In contrast, other closely related strains, such as Limnospira strains (Figure S1-2), which were isolated from ecologically similar but geographically disparate environments (Table 1), have been observed to be distributed by PCA across different metabolomic clusters (Figure 1). One of the strains (B-1526) was located in the close proximity to the Sodalinema strains (Figure 1, Cluster 3.1). Two other Limnospira strains (B-256 and B-287) along with a heterocystous Nodularia strain formed a separate group (Figure 1, Cluster 3.2) that differed from the others by considerable intergroup metabolome variability. This group of strains (B-287, B-256, and B-1529) was distinguished by the highest number of various accumulated metabolites, as demonstrated by the heatmap (Figure 2). Consequently, the strains of the genus Sodalinema exhibit minimal variability in their metabolomes, while strains of the genus Limnospira, despite their close evolutionary relationships, display considerable variability in their metabolomes. This finding might be explained by at least one of the following reasons: 1) A heightened sensitivity of Limnospira strains to the influence of diverse extreme habitat conditions on the formation of strain-specific adaptations, as compared to Sodalinema strains; 2) More variable natural environments in the case of Limnospira than Sodalinema strains, which can reflect evolving more various acclimation strategies in the former; 3) Different capacities of Limnospira and Sodalinema strains to withstand long-term storage cultivation.”

Remark 5: As the authors note, diazotrophs cluster separately in Fig 1 and this is not simply because they have developed similar strategies to cope with an extreme environment – rather, it reflects their ancient genetic legacy. This is acknowledged by the authors in the Discussion (lines 568-577) but the problem this causes for the attribution of metabolites is not discussed specifically. Therefore the clustering analysis cannot tease apart the various potential influences, some of which are of interest for understanding extremophile adaption, and others which are not.

Response: We thank the Reviewer for highlighting the missing part of our results. We agree with the Reviewer's opinion that the diazotrophs cluster separated by PCA in Fig 1 rather reflects their ancient genetic legacy than evolving similar strategies to be adapted to their extreme environments. All six diazotrophic heterocystous strains that were examined in the present study (designated B-1520, B-1519, B-1213, B-1533, B-1535, and B-1529) belong to the order Nostocales, and are characterized by the capacity to form heterocysts for the effective fixation of atmospheric nitrogen. This functionality is likely to be evolutionarily advantageous, as it is retained by cyanobacteria in a variety of extreme environmental conditions. We decided to find patterns of metabolites that might distinguish the heterocystous and non- heterocystous strains and therefore probably be a reflection of alternations in cyanobacteria primary metabolism associated with heterocyst formation for effective nitrogen fixation. Thus, we have added to the Results section a description of pattern of metabolites revealed by comparison of metabolomes of the heterocyst-forming and non-heterocystous strains (lines 496-541):

2.5. Identifying primary metabolomic characteristics associated with heterocyst-forming cyanobacterial strains

and the corresponding Table 6 and Table S2-9 to Supplementary 2 both presenting results of the comparision have been also added. In addition, the Discussion and Conclusion sections have been supplemented with corresponding text to provide a discussion of the results and summarize their outcomes, respectively, and are presented at lines 664-702 and 1111-1115.

Remark 6: As many strategies for extremophile adaption have been described in the literature, pathway analysis can be useful in linking the strain-specific metabolites to known extremophile adaptions, such as compatible solutes. Where this has not been possible, it suggests that new, previously undiscovered adaptions may be present. The trouble is, that such tantalising possibilities are not followed up or validated. Therefore this study throws up several interesting directions for further research, but no hard data or follow up experiments to really support them. The result is indeed an “exploratory” study, a large and unwieldy data repository with many hints but few conclusions that can be drawn.

Response: We thank the Reviewer for his comprehensive evaluation of our study, and we concur with his assertion that this study is of exploratory nature. The objective of this research was to identify the profiles of primary metabolites in strains extracted from markedly divergent environmental conditions but maintained under similar conditions for a long time. These storage conditions involve the selection of media to ensure optimal growth of each strain, whilst concurrently impeding the growth of cyanobacteria through the provision of suboptimal light and CO2 supply conditions. We posit that this experimental design enabled us to accentuate the strain-specific metabolite profiles in stored extremotolerant and extremophilic strains that may serve as indicators of their adaptation strategies to the extreme conditions from which the strains originated. 

We agree with the Reviewer that this study provides “several interesting directions for further research”. Thus, PCA analysis has indicated several noteworthy outcomes relating to the associations between strain phylogenetic closeness and metabolomic variability. These outcomes have been discussed in full at lines 638-663. It is the consensus that the results of this work provide a valuable perspective for at least further research, which we plan to conduct as follows:

 1) The objective of the first planned study is to highlight the differences in metabolomes between cyanobacteria isolated from extreme environments and those cultivated under storage conditions. In addition, the metabolomes of stored cyanobacteria will be compared with those subjected to environmental stress after storage, as is the case in their initial extreme habitat. This investigation will estimate the impact of storage conditions on cyanobacterial metabolism and determine whether their adaptation is sensitive to storage conditions. This investigation is of particular interest when considering groups of terrestrial and freshwater strains, especially those that are tolerant to desiccation and exhibit depleted strain-specific metabolite profiles. 

2) Secondly, the objective is to ascertain the profiles of secondary metabolites in the stored strains studied here. This study will identify the main strain-specific secondary metabolites. Combining this information with the primary metabolomics data obtained here will allow the linkages between primary and secondary pathways to be established and their contribution to the formation of strain-specific adaptations to be assessed. In addition, this study will identify the biotechnological potential of each strain in the production of biologically active compounds.

3) Thirdly, the results of this study highlighted three strains of genus Limnospira which despite of their close evolutionary relationship displayed considerable metabolome variability and Nodularia strain which among all studied strains were distinguished by majority of accumulated compounds. Therefore, these four strains might be a subject worthy of further investigation, specifically with regard to the molecular mechanisms underpinning their high metabolic plasticity.

Accordingly, the Conclusion section has been expanded to include these possible perspectives that can be drawn from the results of this work to further research (lines 1122-1139).

Comment 4: If publication is undertaken, the authors need to, at the very least, draw attention to and address the points raised above. 

Response: Done as suggested and corresponding changes have been added to the revised manuscript. Please see our responses to the abovementioned questions

Remark 7: Similarly, where there is the possibility that key strain- specific metabolites might be contributed by bacterial contaminants, this possibility should be addressed, for example by looking in the literature at known bacterial sources of these metabolites

Response: We thank the Reviewer for this valuable suggestion. The possibility that a trans-isomer of a fatty acid such as trans-9-hexadecenoic acid, which was highlighted in the metabolite profile of Sodalinema orleanskyi (B-2037), could be due to bacterial contaminants has already been indicated in Discussion section (now at lines 838-842 in the revised manuscript). In addition, the corresponding changes with text regarding the strain-specifical metabolites such as salicylic acid, 2C-methyl-D-erythritol, 2C-methyl-D-erythritol-4-phosphate, high abundances of which might also be attributed to the contribution of bacterial contaminants were added in Discussion section and can be seen at lines 745-747 and 803-805 in the revised manuscript.

Corresponding author,

Dr. Tatiana Bilova

Reviewer 2 Report

Comments and Suggestions for Authors

Manuscript ID: ID: ijms-3389849

Title: Probing constitutive traits of metabolic adaptation in cyanobacteria to extreme habitats

The authors investigated the primary metabolites in twelve different strains of cyanobacteria to better understand how they are physiologically adapted to their extreme habitats. The novelty of this research is very weak because this kind of simple/primitive studies is not enough to improve and expand our understanding of the eco-physiological adaptions of these microorganisms to their environments.

The main comment: the primary metabolites might be species-specific, but they are not enough, and not excellent indicators, to conclude or link the ecological and physiological adaptions of these microorganisms. The authors should also assess the secondary and rare metabolites to better understand the mechanisms of their environmental adaptations.

- All the cyanobacterial strains were obtained from the Collection of Microalgae and Cyanobacteria IPPAS. However, the authors should identify them correctly using the polyphasic approach (16S rDNA and ITS sequencing and phylogenetic analyses).

- Data interpretation in the section Discussion is very weak and it should be re-written.

- Names of the cyanobacterial taxa should be in italic in the whole text, e.g. line 117: Nostoc commune and line125: Sodalinema orleanskyi B-2037, S. gerasimenkoae B-353, Limnospira sp., etc.

- Figure 2: It is out of focus.

- The English language should be improved.

Comments on the Quality of English Language

It should be improved 

Author Response

We thank the Reviewer for thoughtful review and highly appreciate the valuable comments and suggestions to improve the manuscript. Following these advices, we have corrected statements and supplemented the manuscript with relevant information in corresponding sections.

Remark 1: The authors investigated the primary metabolites in twelve different strains of cyanobacteria to better understand how they are physiologically adapted to their extreme habitats. The novelty of this research is very weak because this kind of simple/primitive studies is not enough to improve and expand our understanding of the eco-physiological adaptions of these microorganisms to their environments.

Response: We thank the Reviewer for his comprehensive evaluation of our study. However, we do not agree with his assertion that this study presents “kind of simple/primitive studies”. We would like to indicate the exploratory nature of the study that have already been mentioned in the beginning of the Discussion section (a line 611 in the revised manuscript). The objective of this research was to identify the profiles of primary metabolites in strains extracted from markedly divergent environmental conditions but maintained under similar conditions for a long time. These storage conditions involve the selection of media to ensure optimal growth of each strain, whilst concurrently impeding the growth of cyanobacteria through the provision of suboptimal light and CO2 supply conditions. We suggest that this experimental design enabled us to focus on strain-specific profiles of metabolites in stored extremotolerant and extremophilic strains, which may serve as indicators of their adaptation strategies to the extreme conditions from which these strains originate. Thus, this type of exploratory research is regarded as providing hypotheses, interesting ideas for further research, rather than confirming previously set hypotheses. In accordance with this, we once more indicated exploratory character of this study in Introduction and Conclusion sections (lines 110 and 1107). In addition, the Conclusion section has been supplemented with perspectives for further research that can be derived from the results of the present work (see text in lines 1122-1139).

Remark 2: The main comment: the primary metabolites might be species-specific, but they are not enough, and not excellent indicators, to conclude or link the ecological and physiological adaptions of these microorganisms. The authors should also assess the secondary and rare metabolites to better understand the mechanisms of their environmental adaptations.

Response: We thank the Reviewer for arising this question and agree with the Reviewer’s suggestions that ‘secondary and rare metabolites’ might be better indicators than primary metabolites to understand the mechanisms of the cyanobacteria environmental adaptations. However, it was out of score of our study to investigate profiles of secondary metabolites. Instead, we decided to be focus on primary metabolites and the rationale for this was explained in the Introduction section at lines 76-86. We are interested in the primary metabolites because of their involvement in the fundamental metabolic pathways that are essential for life. Consequently, they can be regarded as pivotal mediators of the diversity of adaptive metabolic responses observed in cyanobacteria. While primary metabolites may not possess the same level of specificity as certain secondary metabolites in terms of the development of specific adaptations to a given environment, they function as building blocks or energy compounds for the synthesis of secondary metabolites. Therefore, the profiles of primary metabolites may serve as a valuable guide for the subsequent production of more specialized secondary compounds. Our subsequent objective is to elucidate the profiles of secondary metabolites in the stored strains studied here, with the expectation that this planned study will be able to identify the major strain-specific secondary metabolites. Furthermore, it is planned that combining this secondary metabolite information with the primary metabolomics data obtained here will allow links to be established between primary and secondary pathways and their contribution to strain-specific adaptations to be assessed. Moreover, it is anticipated that this study will unveil the biotechnological potential of each stored strain in the production of bioactive compounds.

Accordingly, in order to clarify the focus of the current study, which is specifically concerned with the primary metabolites of extremotolerant and extremophilic cyanobacterial strains, the title of the study was changed as follows:

“Strain-specific features of primary metabolome characteristic for extremotolerant/extremophilic cyanobacteria under long-term storage”

Remark 3: All the cyanobacterial strains were obtained from the Collection of Microalgae and Cyanobacteria IPPAS. However, the authors should identify them correctly using the polyphasic approach (16S rDNA and ITS sequencing and phylogenetic analyses).

Response: We thank the Reviewer for this valuable suggestion. The strains of genus Sodalinema, IPPAS B-2050, IPPAS B-2037, and IPPAS B-353, are the reference strains of S. stalii, S. orleanskyi and S. gerasimenkoae, respectively, which were described based on a polyphasic approach earlier by Samylina et al. 2021. Other strains were tentatively identified based on phylogenetic analysis of their partial 16S rRNA sequences and morphology. This information has been added to the text of the Result section (lines 117-121) and to the additional Supplementary 1 (Figure S1-1, S1-2)

Samylina, O. S.; Sinetova, M. A.; Kupriyanova, E. V.; Starikov, A. Y.; Sukhacheva, M. V.; Dziuba, M. V.; Tourova, T. P., Ecology and biogeography of the ‘marine Geitlerinema’cluster and a description of Sodalinema orleanskyi sp. nov., Sodalinema gerasimenkoae sp. nov., Sodalinema stali sp. nov. and Baaleninema simplex gen. et sp. nov.(Oscillatoriales, Cyanobacteria). FEMS Microbiology Ecology 2021, 97, (8), fiab104

Remark 4: Data interpretation in the section Discussion is very weak and it should be re-written.

Response: We thank the Reviewer for pointing out the problem. The Discussion section has been revised and enhanced, incorporating a new text that details several findings concerning the associations between phylogenetic closeness between strains and their metabolomic variability (lines 638-663). Additionally, the cluster of diazotrophic heterocystous strains, isolated by PCA has been given particular consideration. Consequently, the Results section has been supplemented with a detailed description of the pattern of metabolites identified through a comparative analysis of the metabolomes of heterocystous and non-heterocystous strains (lines 494-540).

2.5. Identifying primary metabolomic characteristics associated with heterocyst-forming cyanobacterial strains

In addition, the Discussion and Conclusion sections have been supplemented with corresponding text to provide a discussion of the results and summarize their outcomes, respectively, and are presented at lines 664-702 and 1111-1115.

Remark 5: Names of the cyanobacterial taxa should be in italic in the whole text, e.g. line 117: Nostoc commune and line125: Sodalinema orleanskyi B-2037, S. gerasimenkoae B-353, Limnospira sp., etc.

Response: We thank the Reviewer for highlighting this issue. We put all names of the cyanobacterial taxa in italic as suggested throughout the revised manuscript, please see the changes for example at lines 118-138.

Remark 6: Figure 2: It is out of focus.

Response: The quality of the Figure 2 has been improved in the revised manuscript

Remark 7: The English language should be improved.

Response: The manuscript has been also revised and re-worked in order to improve the English language used.

Corresponding author,

Dr. Tatiana Bilova

Round 2

Reviewer 1 Report

Comments and Suggestions for Authors

I was impressed by the closely argued and detailed response from the authors, and the modifications that have introduced to address those points in the manuscript itself. I appreciated reading through the details of their response and while there are still some aspects that I have reservations about, I will accept that the work is exploratory in nature. Their descriptions of the future work they anticipate should address many of these issues, and the paper already is already substantial in size. The statistical treatment is well explained and strengthens confidence in their conclusions, as do the explanations concerning the origins of metabolic similarities between strains. I don't have any further questions to add.  The amended manuscript appears to be acceptable.

Reviewer 2 Report

Comments and Suggestions for Authors

Please change the terms “heterocystes” to “heterocytes”. The same for “heterocystous” and “heterocytous”.

Comments on the Quality of English Language

almost fine.